# A Concept of the Development of Riverside Embankment in the Context of Cracow (A Local Centre)

**Beata Majerska-Pałubicka and Elżbieta Latusek \***

Faculty of Architecture, Silesian University of Technology, Akademicka 7, 44—100 Gliwice, Poland; beata.majerska-palubicka@polsl.pl

**\*** Correspondence: latusek.elzbieta@tlen.pl; Tel.: +48-608-035-396

**Abstract:** The subject of this article is the presentation of site conditions and the authors' concept of the development of the degraded riverside area located in the city of Cracow-Kraków Zabłocie. The concept transforms the above-named area into a multifunctional complex including museum, coworking, business and hotel functions. The area subject to development borders three important districts of Cracow: Old Town (Stare Miasto), Grzegórzki and Podgórze on the bank of the Vistula (Wisła) river. In the land development and urban planning documents of the city of Cracow this area has been marked as the public space which is to become a local focal point or a local centre. The main objective of this work was to find answers to the posed research questions concerning the historic context, formal and legal state, significance for the community as well as economic and ecological implications of the area to be developed. The main purpose was to properly develop the degraded riverside embankment in the downtown environment. The research method was based on own mixed method which encompassed the studies of historical literature and the legal–formal status as well as in situ examinations, including the analyses of the condition of the built and natural environment, traffic and circulation as well as photographic documentation. The authors also familiarised themselves with the activities undertaken by the local community with a view to the area's regeneration. On the grounds of initial investigations, the SWOT analysis was performed and the evaluation of groups of prospective users was conducted. Comparative studies were conducted including selected examples of European riverside development projects. In its assumptions, the proposed concept of the riverside development in Kraków-Zabłocie is to meet the needs of the local community, enable further development of tourism, which is very important to Cracow, and satisfy the paradigm of sustainable development. The effect is a multi-functional complex that becomes an inherent part of the existing context.

**Keywords:** Kraków Zabłocie; Podolski Boulevard; development of riverside embankment; downtown riverside areas; urban local centre; community; historical context; multifunctional complex

## 1. Introduction

Cracow (Kraków), being an important point on the map of Polish historical heritage, is associated mainly with impressive buildings of historic significance well known in Poland and abroad. However, not all city districts have been developing as dynamically as the city centre. Very often the districts of crucial historic importance have been neglected. One of such places was Zabłocie, whose importance has been noticed only in recent years [1]. This area is located in a post-industrial part of the district of Podgórze (Figure 1), in the vicinity of the city's chief arterial roads. There are also two trestle bridges being built at the moment, which are to join the eastern and southern railway exit from Cracow

(Kraków). In addition, there is a newly renovated interchange station Kraków Zabłocie adjacent to the subject-related area from the western direction. Former industrial areas of Zabłocie of some historic importance are being converted into residential areas. As a result, the number of district inhabitants and users has been steadily growing in recent years.

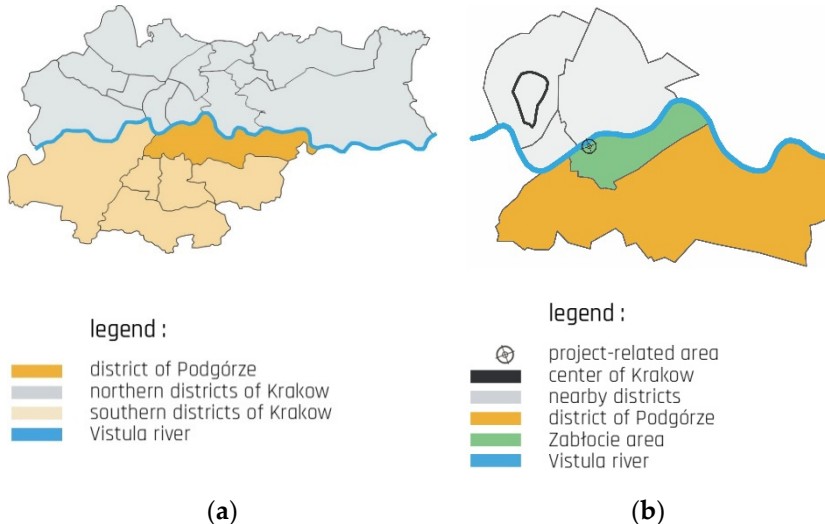

**Figure 1.** Diagrams showing the location of the project-related area: (**a**) Podgórze district area in the context of other Cracow districts; (**b**) Zabłocie area within the scope of Podgórze district and in proximity of the Old Town [2] (elaborated by E. Latusek).

The subject of this work is a fragment of the riverside embankment of the river Vistula (Wisła) in Cracow (Kraków). This riverfront is in many respects a very interesting area, and it has not been properly developed yet. This particular site was selected due to its purpose: in urban planning documents this place is intended for the function of a local centre (Figure 2). It is a socially significant place. This site requires a proper approach to designing with a focus on the creation of architecture. After the research and spatial analyses, it was decided that a multi-functional complex should be designed to meet the needs of Cracow's inhabitants and visiting tourists. This work and investigations concern the area which was designated as a local centre in the Study of Conditions and Directions of Spatial Development of the City of Cracow (Kraków).

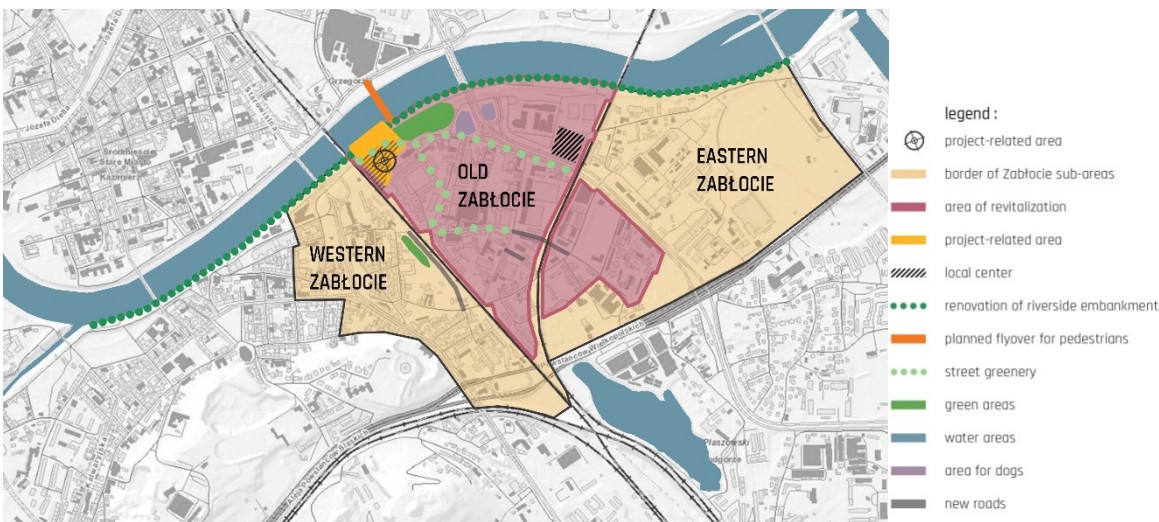

**Figure 2.** Fragment of the regeneration area, sub-area Stare Podgórze–Zabłocie, own elaboration by E. Latusek on the basis of the map extract from the Study.

In 2014, a section of Podolski Boulevard (Podolski Bulwar) was given a new name of Boulevard of the Allied Forces Pilots (Bulwar Lotników Alianckich) because of two important anniversaries: 1 August, the 70th anniversary of the outbreak of the Warsaw Uprising, and 1 September, the 85th anniversary of the outbreak of the Second World War [3]. To commemorate the Polish Air Forces a memorial is planned to be erected there. This fact does not significantly influence the way of development or utilisation of the embankment. Quoting M. Przybyła, "by analysing the course of events connected with the development of the urban space of Zabłocie after 1989, it can be stated that Zabłocie potentially constitutes one of the most important development-prone areas of the city of Cracow" [4].

Everything that is happening around the project-related area shows how interesting and valuable place this is. Two questions arise: Why is this area still neglected? Why has its potential not been used so far? It is worth emphasizing that "one of the crucial, however much delayed, elements of the programme of {Zabłocie regeneration} is the creation of a representative space having functions of a local centre [ ... ] This newly defined showpiece of the area may not only be a magnet attracting tourists interested in the history of technology but also may increase the attractivity of the neighbouring areas" [5].

The main objective of this article is to present methods of the formulation of bespoke concepts of land development in the area of riverside embankment in the downtown zone with a multifunctional complex including the Museum of the Allied Forces Pilots, which will constitute a local centre in the context of the spatial development of a section of the Vistula riverside.

## 2. Materials and Methods

One of the fundamental assumptions of this work is a holistic multifaceted analysis of the study area. To make the most appropriate decisions concerning the selection of a research method, the authors analysed the suitability of a series of research methods usually applied to complex issues requiring an extensive analysis. The study was based on two, in the authors' opinion the most adequate, research methods in a given context:

- Method of logical argumentation—as a search for theoretical interpretation of developments (events) with the application of a logical description of reality, based on analysis and synthesis.
- Heuristic method—understood as ways and rules of proceedings serving the purpose of making the most appropriate decisions in complicated situations, requiring the analysis of available information and the prediction of future phenomena. The method is based on creative thinking and logical combinations [5].

Both the method of logical argumentation and heuristic methods help systemize the activities connected with investigations, beginning from the determination of the study subject and ending at the definition of expected results from the work conducted. In the above-mentioned methods, the subject and scope of the study defined aspects such as the search for key developmental factors, types of investigations (theoretical, analytical), humanistic and philosophical interpretation of architectural issues, etc. Next, the types of undertaken activities are determined (deduction, synthesis, analogies, drawing conclusions, abstract and logical analysis). The heuristic methodology specifies a study approach, for instance, if it is expected to share one's knowledge and opinions, as well as if the researcher should enter some social interactions. Both methods also define techniques applied in order to carry out investigations, such as logical interpretations, SWOT analyses, scenarios, Delphi technique, and foresight. The researcher also determines which tools will be used (architectural documentation, lists and comparisons, and tables). Finally, the expected results are predicted. In the heuristic method, determination of possible directions of development (empirical approach) and definition of methods of implementation of strategic goals (normative approach), or in the case of the method of logical argumentation, for example, if it is planned to publish the study results.

The combination of two methods into one mixed method (Table 1) aims at the systematization of the undertaken work which is supposed to yield specific effects. Although creating their own mixed method, the authors based it on the rules and methods used in deconstructivism, which challenges fixed patterns and ponders on a given issue anew, from scratch, following simultaneously technological changes. This approach was very helpful when it came to the questioning of the legitimacy of the so-far well-established, either by law or custom, decisions, and activities. It facilitated also the search for the balance between individual issues, which had been earlier deconstructed into separate elements. In Tischner's book we read that "Jackques Derrida proposed that the act of creation should be the goal in its own right, not a piece of creation itself, even in architecture. However, this seems to be an extreme view, suspending the centuries-long aim of building engineering" [6].

**Table 1.** Development of own mixed method on the basis of heuristic methods and the method of logical argumentation, (table by E. Latusek, based on [7]).

| Type of Activity | Own Mixed Method |
| --- | --- |
| Subject | A concept of the development of the riverside called Podolski Boulevard (Bulwar Podolski) in Cracow-Kraków Zabłocie into a multifunctional complex. |
| Objective | Development of degraded riverside areas in the downtown zone. |
| Research issue | Proper development of degraded riverside areas in the downtown zone. |
| Thesis | A multifunctional complex including the Museum of Allied Forces Pilots may constitute a local centre in the context of the spatial development of a section of the Vistula riverside embankment. |
| Hypothesis | Will the change of the way of development of downtown riverside areas positively affect the surrounding social environment? |
| Scope of research issues | Theoretical and analytical research, humanistic interpretation of architectural issues, diagnosis of the present-day (social, environmental and economic) state and prediction of directions of development, search for key developmental factors. |
| Activities undertaken | Research on experts' opinions, use of logic, analysis, comparative analysis deduction, synthesis, analogies, drawing conclusions, logical abstract thinking, opinion surveys of local communities and analysis of undertaken local activities of revitalisation |
| Research approach | Sharing knowledge and opinions; search for theoretical interpretation of objective or abstract facts. |
| Techniques applied | Description, explanation, logical interpretation, comparative studies, scale of grades and SWOT analysis. |
| Tools used | Subject literature, architectural and urban planning documentation, computer and software programmes, comparative lists, tables and the Internet. |
| Effects expected | Description of the problem and its interpretation, development of a procedure. Presentation of logical conclusions (academic approach). Prediction of possible ways of development (empirical approach). Determination of ways of implementation of strategic goals (normative approach). |

Similarly to two basic methods, the authors' own mixed method determined the types of activities that had to be undertaken to meet specific investigation needs. The initial phase of the work encompassed the definition of study subject and scope, types of activities, research approach to analyses and design, tools and techniques to be applied as well as expected results. To facilitate the general reception, Table 1 was supplemented with the aspects, such as: subject, work objective(s), research problem, thesis and hypothesis. Own mixed method aimed at the following.

■ Initial determination of the study area resulting in

- ○ historical literature research,
- ○ examination of the formal and legal status,
- ○ analysis of the built and natural environment,
- ○ analysis of traffic and circulation,
- ○ photographic documentation and
- ○ familiarisation with activities undertaken by the local community with the purpose of the area revitalisation.

■ Development of the following items on the basis of the initial research:

- ○ SWOT analysis,
- ○ assessment of the user groups and
- ○ performance of comparative studies with selected European examples of riverside development.

■ Collection and systemization of analysed data according to certain order.
■ Ordering and facilitation of the researchers' activities in the scope of undertaken studies.
■ Facilitation of the researchers' insight into basic issues connected with the conducted analyses.

The research on the formal and legal state of the area of Zabłocie and the project-related land encompassed many urban planning documents, which had an impact on designing decisions in the scope of land development, position of the object on the building plot and its forms. The analysed formal and legal documentation first included documents such as the Building Law, types of flood risk and other legal acts. The legal–formal documentation analysed included first of all documents such as the Act on the Building Law and types of flooding risk. The analysed legal acts concerning the project-related area encompassed the following: A Study of Conditions and Directions of Spatial Development of the City of Cracow (Kraków), Prognosis of Environmental Impact, A Local Programme of Zabłocie Regeneration, Update of Municipal Regeneration Programme of the City of Cracow (Kraków), A Local Plan of the Vistula Riverside "Wisła Boulevards" and A Local Plan of Zabłocie.

*2.1. Research Questions*

- Why has the riverside area of Podolski Boulevard (Bulwar Podolski) not been properly developed yet?
- Will the change in the development of downtown riverside areas positively influence the surrounding community and contribute to the creation of a local centre?

*2.2. Historical Research*

Extensive historical research was conducted beginning with the district of Podgórze, through the area of Zabłocie, Podolski Boulevard (Bulwar Podolski) and Boulevard of the Allied Forces Pilots (Bulwar Lotników Alianckich). This article contains only a fragment of the above-mentioned study due to its length and broad scope. The name "Zabłocie" meant the land is located behind a muddy area, in relation to royal forests "circa Zabloczyc" [4]. The second half of the 19th century was a period of the greatest development of this area. On the eastern side of the town of Podgórze the main railway line of Galicia was constructed. That sparked the development of railway workshops, industrial plants and warehouses as well as a river port. The urban development was steadily increasing and was barred from the Vistula (Wisła) river with a flood embankment. In 1991, Zabłocie was incorporated into District 13-Podgórze. In the early 20th century, many new factories were built there, for instance, Minor Poland's Factory of Enamelware and Metal Products 'Record' (Małopolska Fabryka Naczyń

Emaliowanych i Wyrobów Blaszanych "Rekord"), later Schindler's Factory, today's museum of Oscar Schindler's Enamel Factory. After 1989, the neighbourhood went into decline as a result of liquidation of many state-owned plants and companies. In 1991, an administrative reform merged the areas of Podgórze, Płaszów and historic Zabłocie into the 13th District of the city of Cracow (Kraków) [8].

The list of objects and groups of objects under conservation and legal protection defined by the Municipal Plan of Spatial Development of 2006 clearly pointed out elements that required protection and inclusion in the project. In accordance with the decision on the enlisting of some objects in the project-related area in the heritage register, there are following objects and facilities in "ZONE A–Stare Podgórze (Old Podgórze)" which have been recorded in the heritage register [9]: restaurant "Zabłocie 13" and a cultural and community centre 'Workshop' ('Warsztat') (Figure 3).

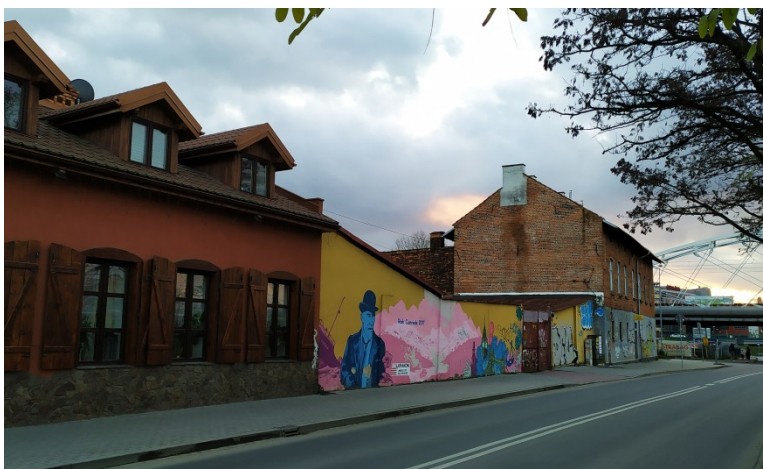

**Figure 3.** Objects recorded in the heritage register (photo by E. Latusek).

## 2.3. Examination of the Contemporary State

Since the late 1990s, activities have been undertaken to regenerate this area, which has been one of the most dynamically developing districts of Cracow. In 2001, Kotlarski Bridge (Most Kotlarski) was built [10], resulting in better transport, traffic and circulation conditions of this part of the city with the city centre. At the beginning of the 21st century, a new educational centre was built in Zabłocie, namely, the Andrzej Frycz Modrzewski Higher School (Krakowska Szkoła Wyższa im. Andrzeja Frycza Modrzewskiego). Also, Cracow Artistic Schools (Krakowskie Szkoły Artystyczne) have been there since 1992. In 2006, Zabłocie was considered to be a strategic area in the development of the city of Cracow (Kraków) and a programme of activisation and regeneration was born [11]. For centuries, the area of Zabłocie played a function of industrial background, initially for the district of Kazimierz, later for Podgórze. Nowadays, there is tendency either to remove or adapt old post-industrial building development into residential objects. Zabłocie is acquiring new features and values thanks to subsequent cultural objects having interesting architectural forms and yet preserving the context of the site. Recent years have witnessed an increase in the number of the area inhabitants. Cultural actions organised by various associations and by the district dwellers are transforming this place into a thriving part of the city. In spite of these changes, the area has not lost its industrial character and is home to a considerable number of small and medium enterprises, printing shops, carpenter's, locksmith's or toolmaker's shops. On the other hand, newly built exclusive apartment buildings (Garden Residence) and modern office buildings (Diamante Plaza) are contributing greatly to the improved and favourable image of Zabłocie (Figure 4).

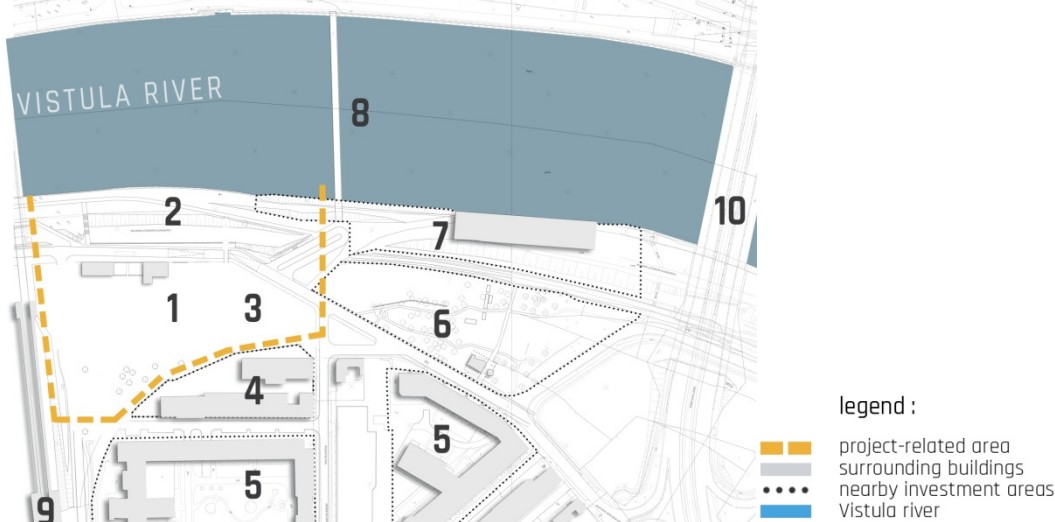

**Figure 4.** Potentially important places in the vicinity of the project-related area: (1) Project-related area; (2) Boulevard of the Allied Forces Pilots (Bulwar Lotników Alianckich); (3) Planned memorial of the Allied Forces Pilots; (4) The project-related area borders with a lighting company; (5) New residential quarters ATAL Residence and Garden Residence; (6) Park "Vistula Station" ("Park Stacja Wisła") which gained an award in the contest for the best developed space in Poland; (7) Planned Cracow Marina ("Marina Kraków"); (8) Planned flyover for pedestrians and bicycles; (9) Railway station Kraków Zabłocie; (10) Kotlarski Bridge (Most Kotlarski) (elaborated by E. Latusek).

The riverside embankment called Podolski Boulevard (Bulwar Podolski), which is the subject of this work, stretches along the right bank of the Vistula (Wisła) river, between the mouth of the Wilga river and the railway bridge in Zabłocie. The embankment is linked with the other bank by means of Józef Piłsudski Bridge (Most Józefa Piłsudskiego) and Silesian Insurgents Bridge (Most Powstańców Śląskich) as well as railway bridges and the flyover for bicycles and pedestrians. The topography of the riverside area is varied. From the direction of the mouth of the Wilga River there stretches a retaining wall built in the late 19th century as flood defence. From the boundaries of the project-related area, the lie of the land changes into a floodbank descending gradually in the direction of the river. The further east from there you go, the more neglected and overgrown with wild plants the landscape becomes. The Local Plan of Spatial Development (Miejscowy Plan Zagospodarowania Przestrzennego) [12] provides for a place where the flyover for bicycles and pedestrians is to be built in order to link two banks of the Vistula (Wisła) river. There is an inland sailing trail on the Vistula River in Cracow called "Waterway of the Upper Vistula River" ("Droga Wodna Górnej Wisły"), which will implement sailing facilities as a further element of the programme of the development of inland waterways in Poland. Prompted by the inhabitants, the authorities of the city of Cracow (Kraków) are planning to implement a project of the Cracow Marina ('Marina Kraków'). There is also an international cycling trail on the route of Cracow–Moravia–Vienna, which is an eco-tourism corridor revealing the cultural, natural and historic heritage of the Central Europe, including the longest alley of trees in Europe.

*2.4. Studies of User Groups*

The research question of whether the change in the ways of development of downtown riverside areas will positively influence the surrounding social environment aims to draw attention to the significance of transformations occurring in the public space. For this reason, the studies of user groups were conducted (Table 2). The Polish society is going to face changes which will reshuffle the labour market; the baby boom generation will retire, and will be replaced by only half the number of new employees. The majority of them will probably be from 'generation Y'. The notion of 'generation Y'

appeared for the first time in 1993 in the magazine 'Advertising Age' denoting the last generation to be born in the 20th century; in Poland it means people born between 1984 and 2000. The expectations of generation Y differ from the expectations of previous generations: "older generations lived to work, whereas millennials work to live" [13]. People at the age of twenty and thirty will dominate the future labour market and because of this the functioning of the working world will depend on their lifestyle, needs and expectations. The characteristic features of the representatives of "generation Y" are as follows:

- they are familiar with technological novelties,
- thanks to the Internet access they live in a "global village",
- they have a less materialistic approach to life than previous generations,
- they are characterised by high self-esteem and high professional competences,
- they are well educated and ready for further development,
- they live longer with their parents delaying the entry into the adulthood and
- they were brought up in the realities of free market.

**Table 2.** User groups in the project-related area and their needs (table by E. Latusek).

| New Residential Objects (Flats and Apartments) | Vistula Boulevards (Bulwary Wisły) | Cracow Academy and Film School | City Centre and the District of Kazimierz | Museums and Cultural Objects | Businesses and Industrial Objects |
| --- | --- | --- | --- | --- | --- |
| New inhabitants (Generation Y) | Tourists and visitors | Students | Tourists and visitors | Tourists and visitors | Employees |
| Children | Cracow inhabitants | Employees of schools and universities | Cracow inhabitants | Visiting school groups | Customers |
| New inhabitants of Cracow after completion of their studies | Customers at riverside restaurants and floating bar barges | | | | |
| | Cycling tourism | | | | |

In combination with negative demographic trends and depopulation of Polish cities, it means a tremendous challenge for local authorities, which will be more and more interested in attracting young specialists in order to maintain their competitiveness [14]. Therefore, the question is not if, but how to compete for young talents? The understanding of the present-day situation of young Poles may provide some suggestions in relation to the project-related area.

Inhabitants living in newly-built residential buildings in Zabłocie will also use and contribute to the local centre. Despite the fact that a local centre was designated within the area of the nearby Cracow Academy College (Akademia Krakowska–Studium), it has no functions either within a wide range of possibilities of community integration or a big choice of entertainment and recreation.

*2.5. Research on the Local Community's Initiatives*

In 2014, with relation to the 70th anniversary of the outbreak of Warsaw Uprising and the 75th anniversary of the outbreak of the Second World War, the City Council passed a resolution to give a section of Podolski Boulevard (Bulwar Podolski) a new name: Boulevard of the Allied Forces Pilots (Bulwar Lotników Alianckich) [15,16]. It was done to commemorate the catastrophe of an Allied Forces' plane "Liberator" [17], which happened in significant historic circumstances, as well as to indicate the connection between this section of the Vistula Valley (Dolina Wisły) with the history of aerial operations over the Minor Poland (Małopolska) region as a result of global political and military developments.

In 1986, thanks to the efforts made by the Cracow Club of Aviation Seniors (Krakowski Klub Seniorów Lotnictwa), the catastrophe was commemorated by placing a memorial plaque in the wall of

the Schindler's Factory at 4 Lipowa Street (ul. Lipowa 4). The plaque pays tribute to the crew of the shot-down aircraft [17].

In 1999, the Institute of Landscaping of the Cracow University of Technology (Instytut Architektury Krajobrazu Politechniki Krakowskiej) drew up a project of the development of the Vistula riverside from the side of Zabłocie. The above-mentioned concept was triggered by the planned re-building of the Vistula river floodbanks. The students' designs referred to, among other things, the creation of a memorial of the catastrophe site and the aircraft crew who met a tragic end. Since 2006, there have been anniversary walks "Liberator above Zabłocie" organised every year on the day of the catastrophe by the Association Podgorze.PL. These walks following the "traces" of the shot-down aircraft Liberator KG-933 attract several dozens of people interested in the past of the city and the district. The walks are advertised in the local and regional media, and the history often appears in the press [18].

Moreover, the local community's initiatives included the creation of a mural showing a Liberator aircraft on the wall of the building located at 14 Dąbrowskiego street (ul. Dąbrowskiego 14) in Cracow (Kraków) as well as the raising of a memorial obelisk at the exit of Przemysłowa street (ul. Przemysłowa) to commemorate the site of the catastrophe.

The professors and students of the Faculty of Landscaping of the Cracow University of Technology undertook the task of preparing the land development project and the concept of the memorial as continuation of the efforts of the aviation and pilot community aiming to commemorate the aerial combat over Cracow (Kraków) and Minor Poland (Małopolska) region [19].

## 2.6. Legal and Formal State

The Local Plan of Spatial Development 'Zabłocie' [12] prepared in 2006 is no longer up to date due to recent transformations the area of Zabłocie is undergoing. The discussed Podolski Boulevard (Bulwar Podolski) stretches between the mouth of the Wilga River, which flows into the Vistula and a railway bridge (Figure 5). The Local Plan of Spatial Development "Vistula Boulevards" ("Bulwary Wisły") (2013) [20] defines the rules and regulations of space formation, but only does so for the western part of the riverside area. In the document Study of Conditions and Directions of Spatial Development (2014), an analysis of the justification of the preparation of a new plan Zabłocie–East (Zabłocie–Wschód) was made, which demonstrates the Cracow City Council's interest in further development of this post-industrial part of Podgórze.

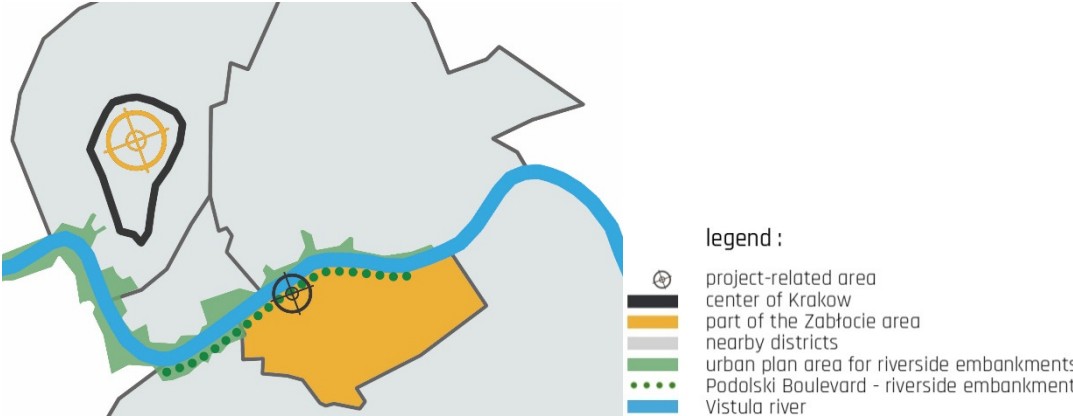

legend :
⊕  project-related area
  center of Krakow
  part of the Zabłocie area
  nearby districts
  urban plan area for riverside embankments
•••• Podolski Boulevard - riverside embankment
  Vistula river

**Figure 5.** Boulevard of the Allied Forces Pilots (Bulwar Lotników Alianckich) within Podolski Boulevard (Bulwar Podolski) only partially covered by the Local Plan of Spatial Development of the area called 'Vistula Boulevards' ('Bulwary Wisły') [20], (elaborated by E. Latusek).

## 2.7. Research on Urban Development Composition

Both national and European roads run through Cracow (Kraków) (Figure 6). Typical traffic intensity during rush hours does not exceed critical limits. High traffic intensity occurs along the

second ring road of Cracow, in Gustawa Herlinga-Grudzińskiego Street and in the vicinity of the Kotlarski Bridge and the Marshal Józef Piłsudski Bridge. However, traffic jams appear on the regional road no. 776 in Powstańców Wielkopolskich Street.

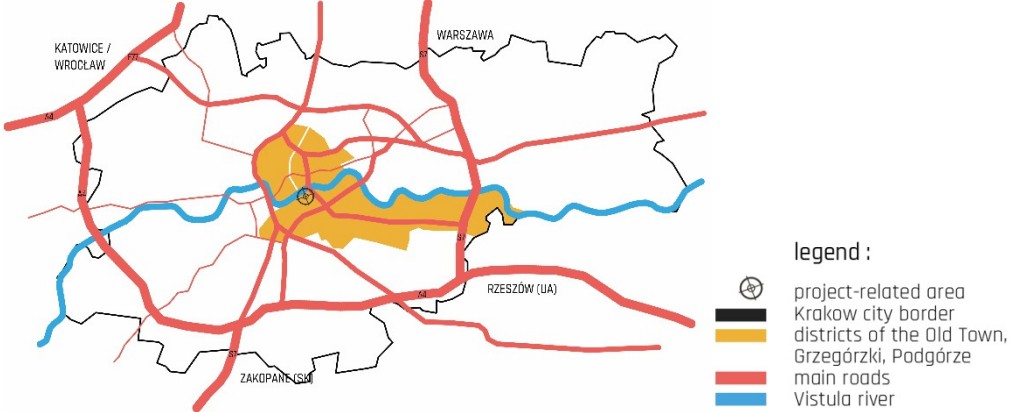

**Figure 6.** Main arterial roads of Cracow (Kraków), (elaborated by E. Latusek).

Cracow (Kraków) is one of the largest railway interchange stations in Poland (Figure 7). It is linked to the majority of cities in Poland, including express Pendolino links with Warsaw (Warszawa) and Gdańsk. In addition, it has international connections with Vienna, Prague, Budapest and Lviv. The Main Railway Station in Cracow along with the Małopolska Region Coach Station, municipal public transport (buses, underground fast tram) and the link to the Cracow-Balice International Airport make up a complex called the Cracow Public Transport Centre. By the end of 2020, four new rail tracks will have been built on two newly constructed railway trestle bridges on the crosstown line. The Polish Railways PKP Polskie Linie Kolejowe S.A. (Joint Stock Company) link the central railway station with the station Kraków-Płaszów facilitating thus the traffic of agglomeration and long-distance trains. The station Kraków-Zabłocie is currently under modernization, which is connected with the above-mentioned investment.

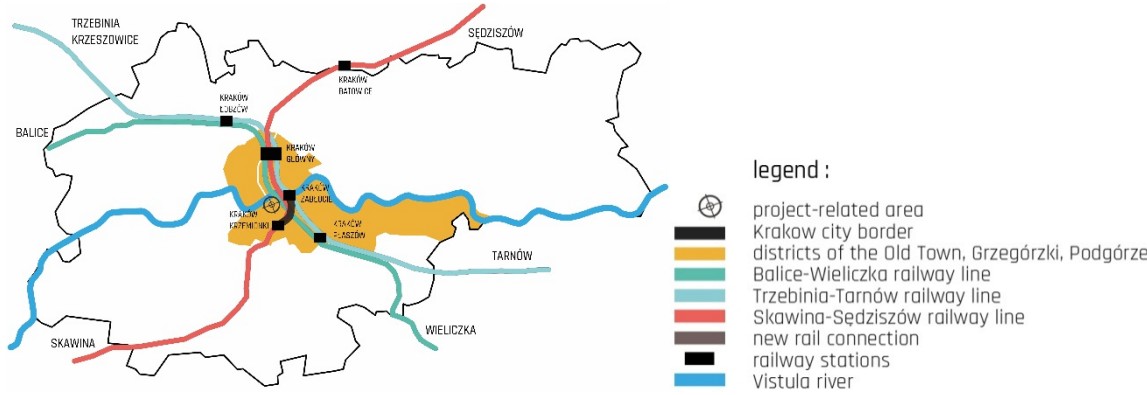

**Figure 7.** Railways in Cracow (Kraków), (elaborated by E. Latusek).

Poland is to ultimately house five green bicycle trails referred to as greenways. Local bicycle loops have been opened on the Amber Greenway (Szlak Bursztynowy: Budapest-Bańska Szczawnica-Cracow-Gdańsk). There is an international trail between Cracow–Moravia–Vienna, being an eco-touristic corridor exhibiting the cultural, natural and historic heritage of Central Europe. In the future, the aforesaid corridor should become the longest "alley of trees" in Europe. In the direct vicinity of the study area there are many local bicycle lanes and a public bike rental system.

Along the river Vistula (Wisła) in Kraków there is an inland shipping route known as "Waterway of the Upper Vistula River" (Droga Wodna Górnej Wisły). In 2018, the Ministry of Marine Economy

and Inland Navigation signed an agreement for the development of a transport analysis, being the first study of this type in relation to inland water transport. The analysis should concern inland navigation on the river Odra and Wisła, as another element of the programme aimed to develop inland waterways in Poland. The city of Cracow, encouraged by its residents, is planning to implement a project named "Marina Kraków".

### 2.8. Urban Development Dominants

Near the study area there are three new high-standard residential complexes. South of the area there are spaces with strong historical connotations: Cricoteka, Ghetto Heroes Square, a concept to create the Planet Lem object, Oscar Schindler's Enamel Factory and Museum of Contemporary Art in Cracow-MOCAK (Figure 8). Nearby large educational establishments include the Andrzej Frycz Modrzewski Higher School in Cracow, the Institute of Ceramics and Building Materials, Glass and Building Materials Division in Kraków-Podgórze, AMA Film Academy, students' dormitory of the Academy of Music in Cracow and the Adam Mickiewicz Secondary School of General Education no. 4. Nearby hotels include 4-star standard Qubus Hotel, Hotel Galaxy, PURO Hotel Kraków Kazimierz and INX Design Hotel as well as many other hotels located in the district of Kazimierz. Nearby recreational facilities include shopping mall Galeria Kazimierz, Saturn Fitness, Gym Park fitness centre, FitNOW fitness centre and dietician's, Laserpark laser entertainment centre and, located by the river, Wisła: a water tram stop and a kayak rental point. On the opposite bank of the river Wisła there is Galeria Kazimierz shopping mall, which, in the future, will be accessible via a footbridge (for pedestrians and cyclists).

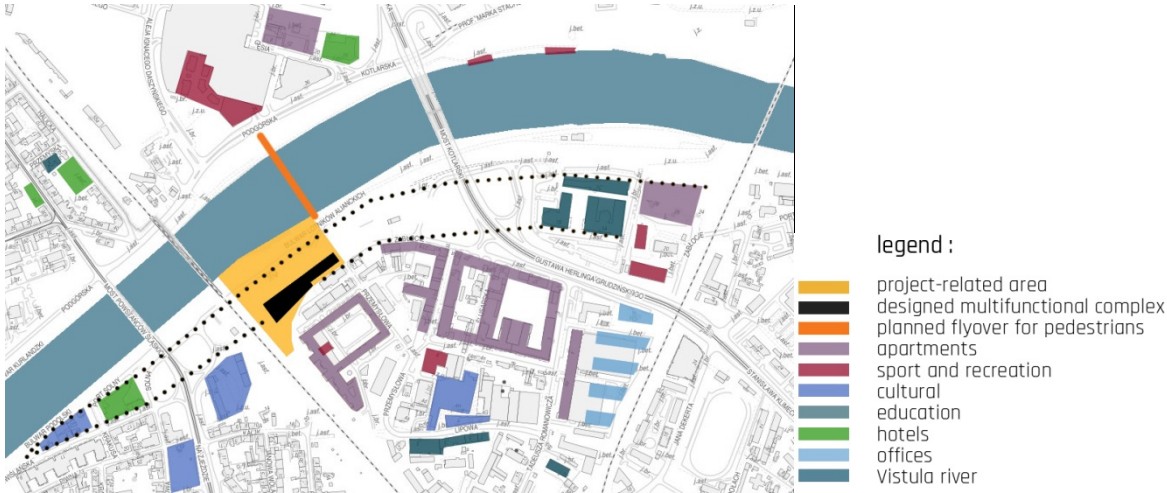

**Figure 8.** Significant characteristic building development surrounding the study area (elaborated by E. Latusek).

### 2.9. Major Vistas

The area of Podolski Boulevard (Bulwar Podolski) is located between the mouth of the Wilga river and the railway bridge in Zabłocie in the district of Podgórze. The Local Development Plan for the Area of the Vistula Riverside, the so-called "Wisła Boulevards", contains regulations related to land development, yet only in relation to the western part of Podolski Boulevard, without its eastern part located in Zabłocie (Figure 9). This part of riverside including areas located east of the railway bridge has not been regulated in terms of land development, supplementation of landscape architecture and lighting, adjustment of greenery. The general plan provides for related supplementation as well as the maintaining of main passageways and viewpoints.

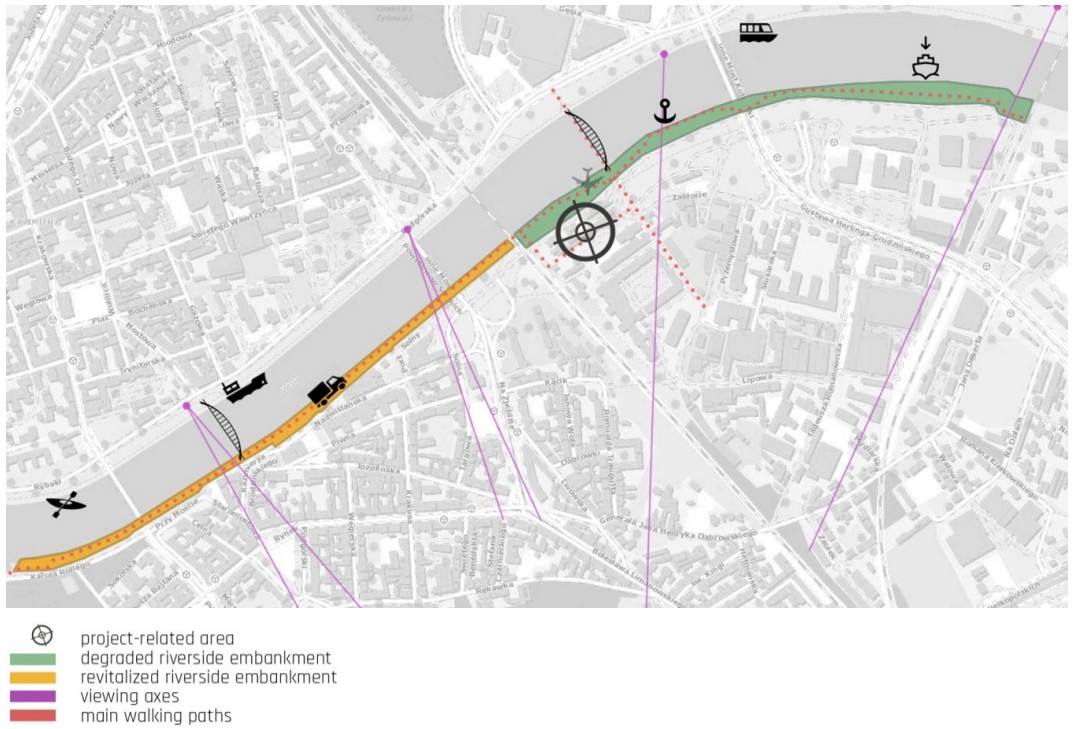

project-related area
degraded riverside embankment
revitalized riverside embankment
viewing axes
main walking paths

**Figure 9.** Main vistas in the context of Podolski Boulevard (Bulwar Podolski) (elaborated by E. Latusek).

### 2.10. Research on Greenery Elements

The preliminary analyses related to the study area provoked a number of questions, one of which is concerned with the lack of the appropriate development plan for Podolski Boulevard. The detailed assessment of the existing condition revealed that an intended space of historical commemoration was to be the Boulevard of Allied Forces Pilots (Bulwar Lotników Alianckich). Today, this area is still wasteland (Figure 10) (overgrown with grass and high greenery) despite the fact that nearby there are new apartment buildings and the Andrzej Frycz Modrzewski Higher School in Cracow. Only a plaque with the boulevard name stresses the significance of the area. Because of the fact that Zabłocie is characterised by the high-density housing development of the city centre, and yet does not have a payable parking zone, today the boulevard is often used as a "wild" car park.

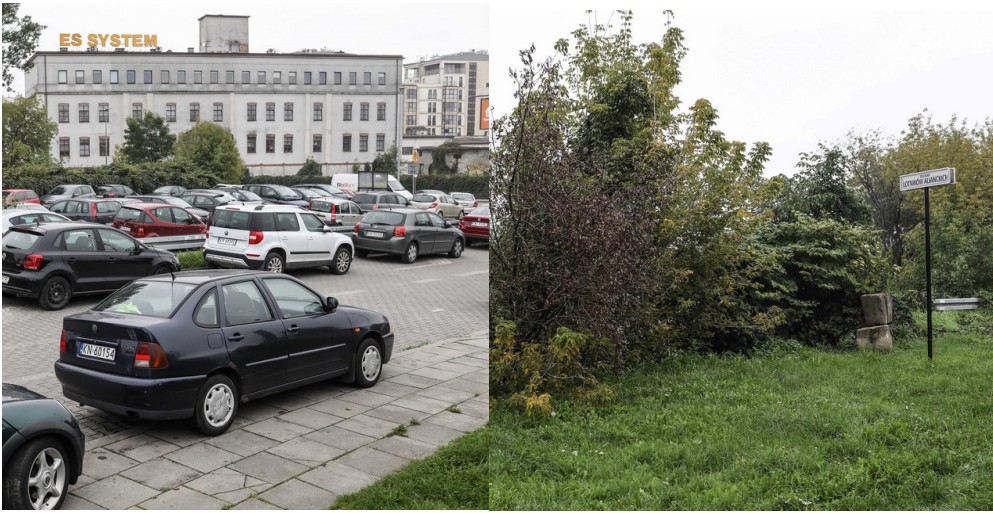

**Figure 10.** Plots constituting the main part of the development of the design of a multi-functional complex with objects and sites under the conservation (photo by E. Latusek).

### 2.11. Architectural Research

Historic housing development of the district of Podgórze is undergoing gradual regeneration (Figure 11). In the area of post-industrial Zabłocie, there is one of the most important historic objects, namely, Oskar Schindler's Enamel Factory (Figure 12). The owner of the factory rescued Jews from the Holocaust during the Second World War, which was shown in the film "Schindler's List" made in 1993. The Municipal District Authorities incorporated this building into the Historical Museum of the City of Cracow (Muzeum Historyczne Miasta Krakowa) in 2005. The permanent exhibition held in this place "Cracow–Under Occupation 1939–1945" received an award for the best historical exhibition in Poland in the contest 'Sybilla 2010'. Monthly, as many as 15–20 thousand tourists from all over the world visit this exhibition [21].

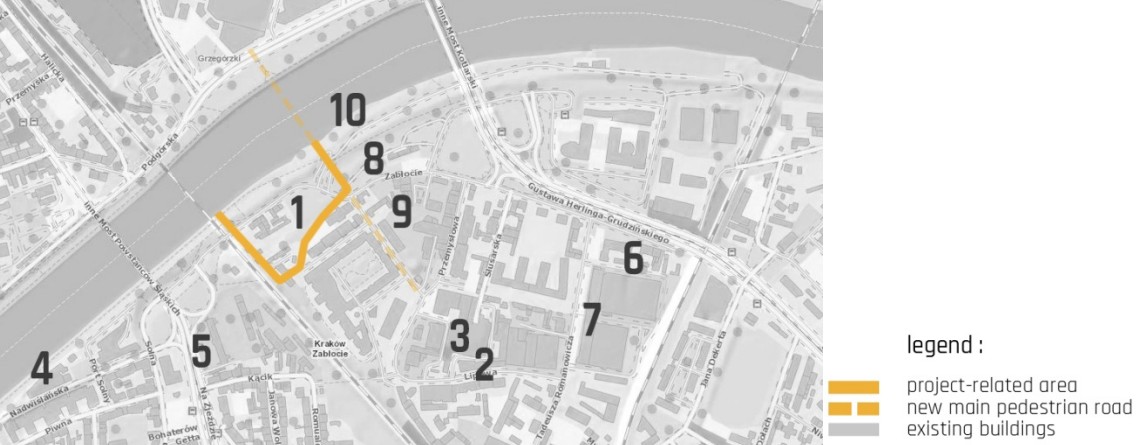

**Figure 11.** Cultural dominants of Podolski Boulevard (Bulwar Podolski): (1) Project-related area; (2) Oskar Schindler's Enamel Factory; (3) Museum of Contemporary Art in Kraków-MOCAK; (4) Centre for the Documentation of the Art of Tadeusz Kantor–Cricoteka; (5) Centre of Literature and Language–'Planet Lem' ('Planeta Lem'); (6) Zabłocie Business Park; (7) Student Hall of Residence Livinn Cracow; (8) Park 'Vistula Station' ('Stacja Wisła'); (9) ATAL Residence; (10) Cracow Marina, (elaborated by E. Latusek).

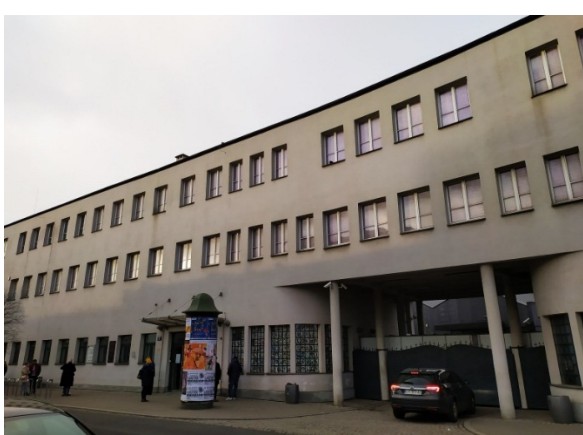

**Figure 12.** Museum-Oskar Schindler's Enamel Factory in Cracow (Item 10, Figure 1) (photo by E. Latusek).

In contrast to the heavily historically-oriented Oskar Schindler's Enamel Factory, there is the Museum of Contemporary Art in Kraków (MOCAK) (Muzeum Sztuki Współczesnej w Krakowie) (Figure 13) focusing on ethical and cognitive values, and showing the connection art has with everyday life. The exhibitions encompass the latest international art, education as well as research and publishing projects.

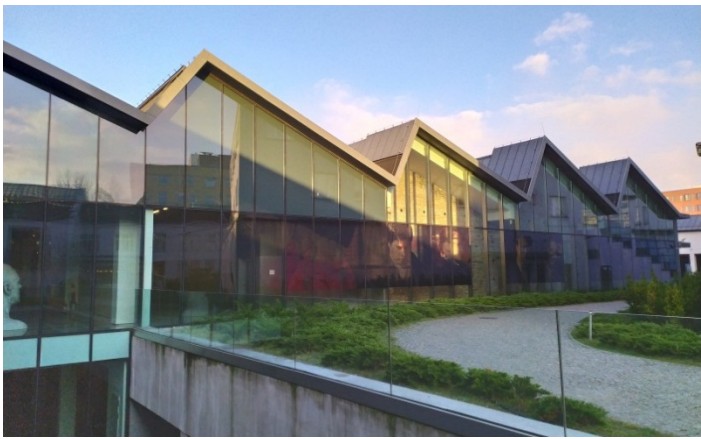

**Figure 13.** Museum of Contemporary Art in Kraków-MOCAK, designed by Claudio Nardi Architette (Item 10, Figure 1) (photo by E. Latusek).

Another characteristic building facing the Vistula (Wisła) river consists of the facilities of the former Podgórze Power Plant and plays a function of the Centre for the Documentation of the Art of Tadeusz Kantor-Cricoteka (Centrum Dokumentacji Sztuki Tadeusza Kantora–Cricoteka) (Figure 14) [22]. The form of regeneration of the former power plant presented by designers (Architectural Office Vision-Biuro Architektoniczne Wizja and nsMoonStudio) represents an interesting way of activisation of the Vistula (Wisła) riverside.

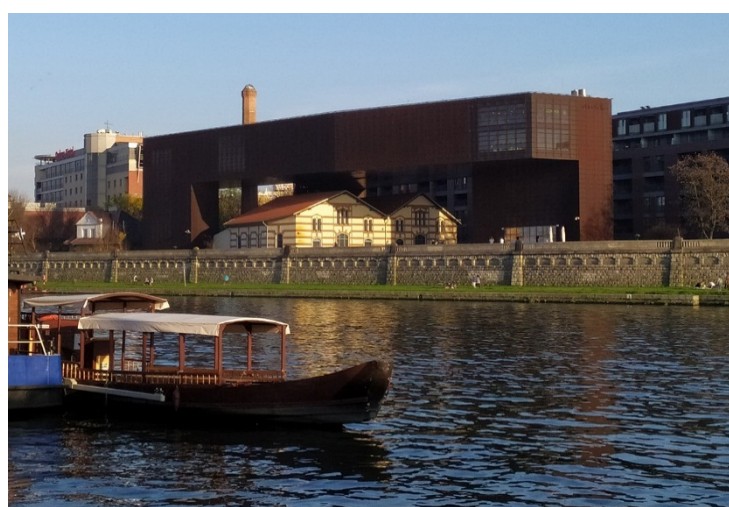

**Figure 14.** Centre of the Documentation of the Art of Tadeusz Kantor-Cricoteka, designed by Biuro Architektoniczne Wizja and nsMoonStudio (Item 10, Figure 1), (photo by E. Latusek).

In March 2019, an architectural and urban planning competition was adjudicated. It aimed to select the best concept of a multi-functional centre of literature and language. The winning concept plans to regenerate the 18th century Salt Warehouse (Skład Solny) located at 8 Na Zjeździe street (ul. Na Zjeździe 8) in Cracow and the creation of the Centre of Literature and Language–Planet Lem (Centrum Literatury i Języka–Planeta Lem). The object is supposed to become an operational centre for the programme Cracow City of Literature (Kraków Miasto Literatury) UNESCO [23].

Zabłocie being one of the oldest Cracow districts lies in the vicinity of the city centre, a fashionable district of Kazimierz and numerous colleges, which undoubtedly is a great asset from a perspective of the localisation of office buildings. On the site of the Cracow Electronic Plants Telpod (Krakowskie Zakłady Elektroniczne Telpod) including shop floors of approximately 5 000 m$^2$ and 10 000 m$^2$, a new office complex Zabłocie Business Park is to be created [24]. The first 7-storey building of class A, having

a BREEAM certificate and offering 11 300 m$^2$ of surface, was commissioned in 2017. Another, out of four objects, will be completed in mid-2020 [25]. In the same area, a student residence hall, Akademik Livinn Kraków, was built (Figure 15). The building includes 290 flats for over 700 students.

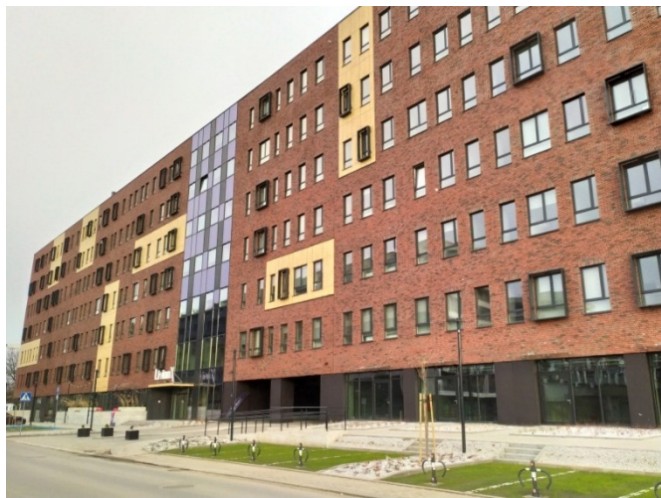

**Figure 15.** Student Residence Hall Livinn Kraków, renovation and modernisation by Unibep (Item 10, Figure 1) (photo by E. Latusek).

In June 2017, in the area of the former railway station Podgórze-Wisła (later Kraków-Wisła) a park was created called Park Vistula Station (Park Stacja Wisła) (Figure 16). The competition was won by the design devised by Michał Grzybowski, a post-graduate student of the Landscaping at the Cracow University of Technology (Architektura Krajobrazu, Politechnika Krakowska). The park obtained an award in the competition for the best developed space in Poland [26]. The nearby premises of the former factory "Miraculum" were earmarked to be the site of construction of high-class residential buildings ATAL Residence (Figure 16) [27]. The investment was implemented nearby the Vistula (Wisła) river in Zabłocie street. A character of this building development fuses modern and post-industrial features, matching a new face of Zabłocie to its historic landscape. This policy contributes to the fact that Zabłocie is one of the best developing parts of Cracow (Kraków).

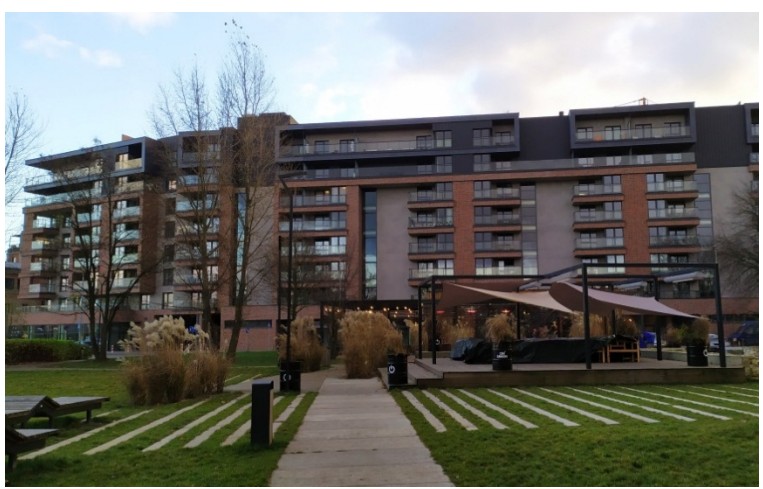

**Figure 16.** Park Vistula Station (Park Stacja Wisła), designed by Michał Grzybowski. In the background, residential building ATAL Residence Kraków, designed by Biuro Rozwoju Krakowa S.A. (Item 10, Figure 1), (photo by E. Latusek).

Not far from the project-related area, in the vicinity of Kotlarski Bridge (Most Kotlarski), a new investment is being planned, namely, Cracow Marina (Marina Kraków). Due to its location in the city centre, the marina will attract tourists. It will also become a leisure and recreation centre for the Cracow inhabitants who want to spend their free time by the river. The marina facilities are planned to stretch over a distance of approximately 1 km [28].

*2.12. SWOT Analysis*

The SWOT analysis (Table 3) performed in this work summarises the conducted investigations and classifies all information obtained to identify assets and advantages of the space analysed as well as weaknesses showing design barriers. The analysis also shows opportunities and aspects bringing benefits for the area analysed. On the other hand, it reveals threats and risks connected with unfavourable changes.

**Table 3.** SWOT analysis concerning the conditions of the existing project-related area (table by E. Latusek).

| Strengths | Weaknesses |
|---|---|
| 1. Historic development of the district of Podgórze is undergoing gradual regeneration. | 1. Post-industrial areas are neglected, littered and serve as wild parking spaces. |
| 2. Local community's activities connected with the commemoration of the Liberator aircraft catastrophe. | 2. Plan to combine the investment of the flyover for bicycles and pedestrians with the construction of the memorial delays their implementation. |
| 3. Diverse topography of the riverside embankment (a vertical wall of flood defences vs. floodbank gradually going down towards the river) as attractive elements of the development. | 3. Diverse topography of the riverside (a vertical wall of flood defences vs. floodbank gradually going down towards the river) as a design-related impediment and a cost-increasing factor. |
| 4. Main traffic and transport arteries in the vicinity. | 4. Increased traffic. |
| 5. Podolski Boulevard was one of the main topics in the political campaign in the election of local self-governments in 2018. | 5. Declarations made during the election campaign were not put into practice and implemented. |
| 6. Attempts made to regenerate this part of Cracow proved very successful (cultural events, new cultural objects); at the same time, industrial character of the area was preserved. | 6. Many historic industrial objects were demolished (for instance, Miraculum factory, Telpod industrial plants). |
| 7. Tendency to remove or adapt old buildings for the purpose of multi-family residential buildings resulting in the increase in the district population. | 7. Difficulties resulting from joining different plots of land. |
| **Opportunities** | **Risks** |
| 1. Area borders three important districts: Kazimierz, Podgórze and Grzegórzki. | 1. Traffic jams during rush hours. |
| 2. Area features objects under conservation. | 2. Impediments in the transformation process of land development. |
| 3. Railway station Kraków Zabłocie was renovated, which may decrease traffic. | 3. Noise made by trains will become difficult for the users of offices and residential buildings. |
| 4. Two railway trestle bridges will be built. | 4. Degradation of the environment of the ecological corridor of the Vistula (Wisła) river. |
| 5. Coordination of the circulation routes with the planned cycling and pedestrian flyover and cycling paths along the Vistula (Wisła) riverside. | 5. Lack of proper separation of the circulation of pedestrians, bicycles and cars may lead to traffic and transport difficulties. |

**Table 3.** *Cont.*

| | |
|---|---|
| 6. Possibility of using the EU funds for the recultivation of a part of the riverside situated in a good location. | 6. Necessity of meeting restrictive requirements of the EU projects. |
| 7. A new Local Plan of Land Development is going to be devised. | 7. Introduction of big changes in the district development without an updated Local Plan. |
| 8. Enrichment of the embankment with public facilities having an interesting architectural form may positively influence the image of Cracow from the riverside. | 8. Enrichment of the embankment with public facilities may unfavourably influence the natural environment of the Vistula riverside. |
| 9. Tendency to remove or adapt old buildings for the purpose of multi-family residential buildings resulting in the increase in the district population. | 9. Necessity of the removal of derelict or temporary buildings; flood risk. |
| 10. A local centre for the integration of the local community was provided for in the Study of Conditions and Directions of Spatial Development of the City. | 10. Conflict of interests between local community, investors, city authorities and environmental requirements. |
| 11. Further land development should match the award winning public space of the Park Vistula Station (Park Stacja Wisła) and the concept of Cracow Marina (Marina Kraków). | 11. Spatial and functional conflicts. |

## 3. Conclusions Based on Investigations

The strategic analysis showed that the most of the issues connected with the strengths of the existing project-related area, as well as opportunities of its development, may result in the transformation of this area into well-functioning space with regard to spatial, economic, social and cultural aspects. On the other hand, there are weaknesses which result from long-term negligence. However, nowadays efforts are made to eliminate them. The risks can be eliminated already in the initial phase of designing new spatial development.

Necessity arises of the development of Boulevard of the Allied Forces Pilots (Bulwar Lotników Alianckich) and creation of an attractive multi-functional space, which would serve as a local centre attracting local residents, tourists and visitors. A museum space connected with the history of the Allied Forces Pilots should constitute a memorial commemorating bygone events and at the same time be an important element of this area. Other functions, such as offices, hotels and recreation facilities should be gradually completed. The designed development should constitute the continuation of attractive public facilities facing the Vistula (Wisła) river.

*Guidelines*

In the Study of Conditions and Directions of Spatial Development of the City of Cracow (Kraków), the analysed area is located within the boundaries of two structural urban planning units. The delineated roads are to correspond to the guidelines provided in the local development plans. The above-mentioned guidelines for the local development plans define also the category of the analysed area as the services areas (Figure 17). Their primary purpose is to become the building development intended for the following functions: offices, culture and other services along with some necessary ancillary buildings and accompanying greenery. The optional function includes arranged or unarranged green spaces, such as parks, squares, greenery spots, river parks as well as green belts around buildings and vegetation screens (the so-called "vegetation barriers").

The Resolution of the Council of the City of Cracow (Kraków) of 28 June 2006 on Passing the Local Plan of Spatial Development of the Zabłocie Area general regulations constitute that the existing building development should be preserved or rebuilt, new building development should be implemented and new investments made, the changes in the use and development of the land should be introduced. The existing valuable buildings and areas can be utilised in the previous way, until these areas are newly developed according to plan. The whole area covered by the plan is located on the

terrain where there is a risk of landslide. No admissible noise levels in the environment were defined for this area in the development plan. The space of the site needs putting in order, that is, the integration of land plots and removal of some derelict buildings. The objects under conservation, namely, a restaurant (Zabłocie 13) and a culture and community centre 'Workshop' ('Warsztat') (Zabłocie 25) may be included in the development context as objects of historical significance.

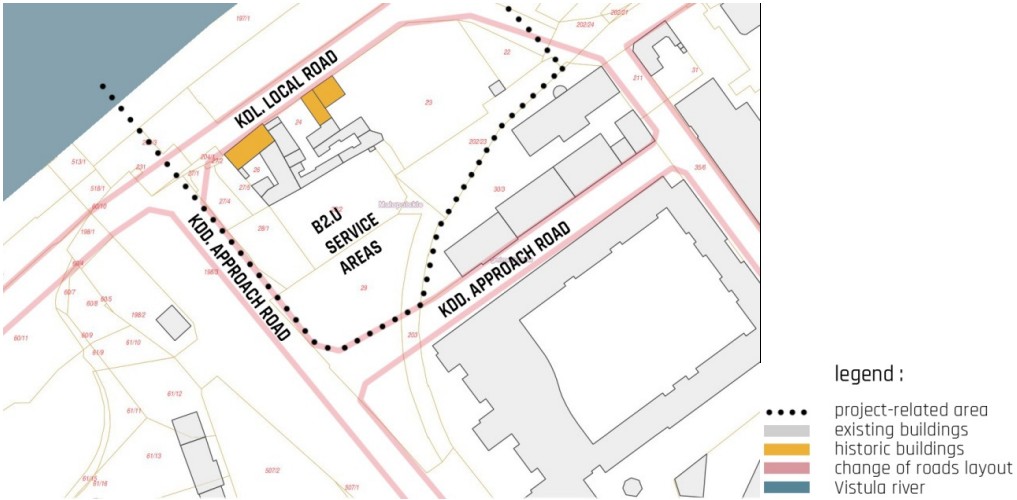

**Figure 17.** Plots constituting the main part of the development of the concept of a multi-functional complex with objects and sites under the conservation (elaborated by E. Latusek).

## 4. Discussion/Proposed Solutions

The planned development of the project-related area located in Zabłocie Street is to provide answers to the above-posed research question: Whether the change in the development of downtown riverside areas will positively influence the surrounding community and contribute to the creation of a local centre? A newly designed multifunctional complex will fill in the space in the existing buildings providing services and having representative functions (Cricoteka, Qubus Hotel, Cracow Academy-Akademia Krakowska) (Figure 18). The conditions for the localisation of the new object are that it must have a supplementary character matching the existing building development, it must comply with the requirements of the Local Plan and must not cross the boundaries of the building development marked in the drawing of the local site plan.

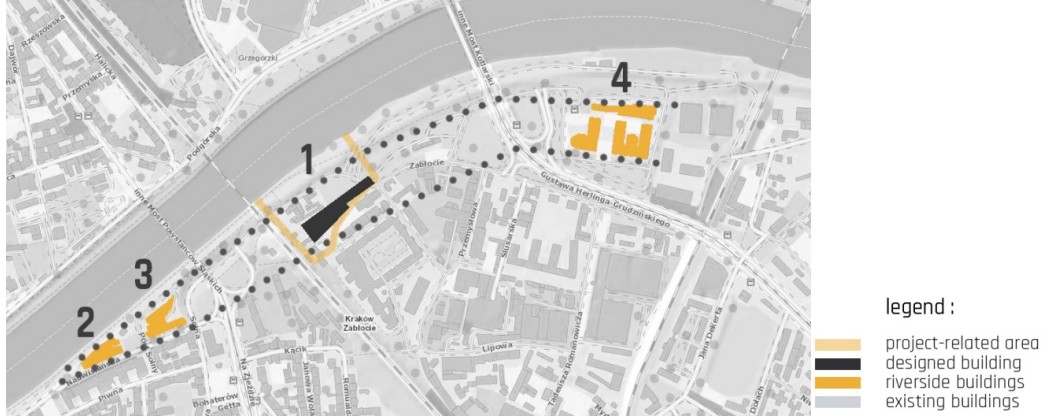

**Figure 18.** Design concept in the context of surrounding building development: (1) Project-related area; (2) Crikoteka; (2) Qubus Hotel; (3) Cracow Academy (elaborated by E. Latusek).

### 4.1. Greenery

The development of the document "Trends of the Development and the Management of Green Areas in Kraków in the Years 2017–2030" involved the development of the electronic space-related "Concept of the Public Green Areas System" and "Register of Green Areas", helpful in the continuous management and maintenance of the areas of greenery by the Department for Municipal Greenery Management. As can be seen in the diagram below (Figure 19), a large concentration of greenery is located in the Park 'The Vistula Station' (Park Stacja Wisła). Along the Podolski Boulevard there are single groups of trees. Only the area designated for the "Marina Kraków" investment is characterised by the greater density of riverside greenery and high trees. The study area is intended to be planted with many trees and shrubs. The study area will be maintained as the area of ecological and landscape greenery. Ecological aspects of the introduced changes are as follows.

■ In accordance with the directions of development and the management of green areas in Cracow (Kraków), the green character of the study area has been maintained in the form of a green area of significant landscape and ecological values.
■ It is planned to plant a large number of park trees and bushes in the study area as well as to implement small architecture and lighting in the scope required for the safety of use.

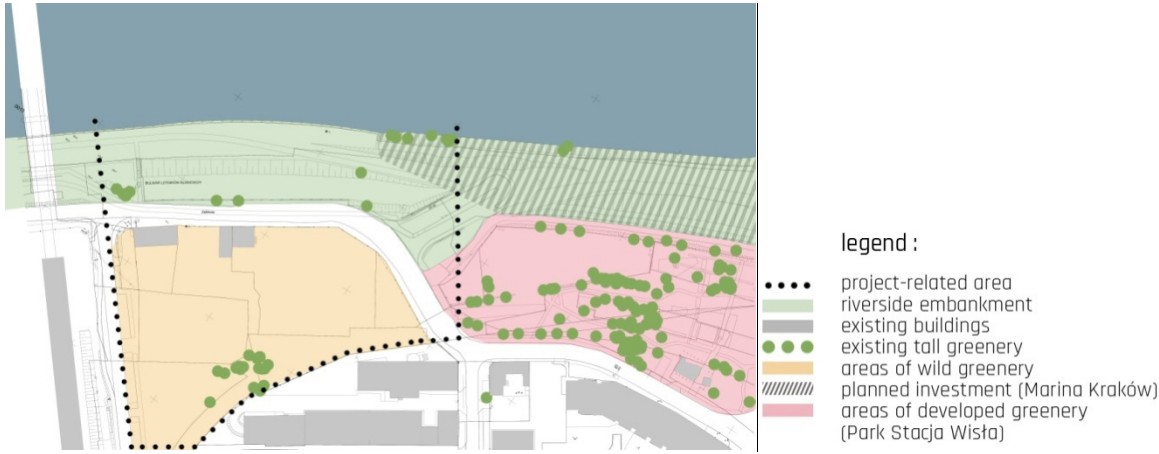

**Figure 19.** Analysis of greenery, (elaborated by E. Latusek).

### 4.2. Cubature Buildings

The designed multifunctional complex is located between historic buildings (from the northern direction) and buildings belonging to the lighting company (from the southern direction). The main limitations to the surface of the building development are as follows (Figure 20):

1.  Areas of a potential flood risk (from the north). The outline of the building was designed in such a way so as not to cross the potential line of flooding in case the Vistula bursts its banks. For this reason, the surface of the planned building development amounts to ~25% of the whole project-related plot. A vast area in front of the designed building may practically play a function of riverside boulevards and constitute the local centre, or be a continuation of an attractive space of the Park Vistula Station "Park Stacja Wisła".
2.  The proximity of the building complex, as close as possible, to the boundary of the plot and neighbouring objects belongs to the lighting company (from the south).

The design preserves the existing historic buildings located in the project-related area (Workshop "Warsztat" and "Zabłocie 13") and provides for a small cubature building, which is to serve as sports equipment rental, in their immediate vicinity.

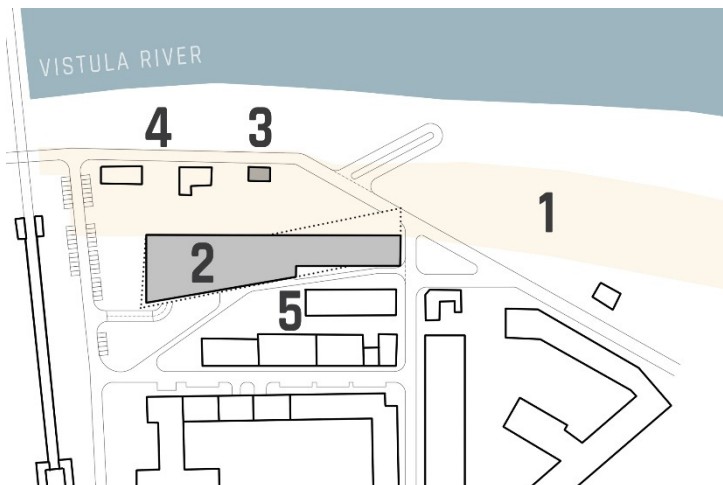

**Figure 20.** Embedding the object in the context: (1) Flood risk area, (2) Designed multifunctional complex, (3) Designed sports equipment rental, (4) Existing historic buildings, (5) Existing objects of the lighting company (elaborated by E. Latusek).

*4.3. Transport, Traffic and Circulation*

The design provides for the coordination of circulation routes with the planned cycling and pedestrian flyover as well as cycling paths along the Vistula (Wisła) riverside (Figure 21). The pedestrians' path (east–west) will facilitate the circulation of pedestrians to and from the existing railway station Kraków Zabłocie. A fragment of the main street ul. Zabłocie (east–west) was lowered, which enabled positioning of a smaller pedestrian flyover over the road. It is most probable that the traffic in Zabłocie street will be significantly increasing, therefore thanks to the flyover pedestrians and cyclists will be able to safely circulate between the designed multifunctional complex and the existing green areas located on the Vistula embankments. The designed underground car park with 70 places and newly-marked parking spaces located along the access road to the planned building development will prevent the creation of wild parking spaces in the riverside area. Ecological aspects of the introduced changes are as follows.

■　Reduction of the $CO_2$ emission due to the restriction of motor traffic in favour of cycling.
■　Expansion of biologically active areas thanks to the design of an underground car park and parking spaces along the access roads.
■　Concept of a new footbridge for pedestrians and cyclists (a cycling link between the districts of Grzegórzki-Podgórze) to improve, in an ecological way, the accessibility of the area for new users coming from outside the district.

A social aspect is as follows:

■　Increase in the safety of users by separation of motor traffic from pedestrians and cyclists thanks to a pedestrian and cycling footbridge over the street of Zabłocie, linking the study area with the adjacent areas of the Boulevard of the Allied Forces Pilots.

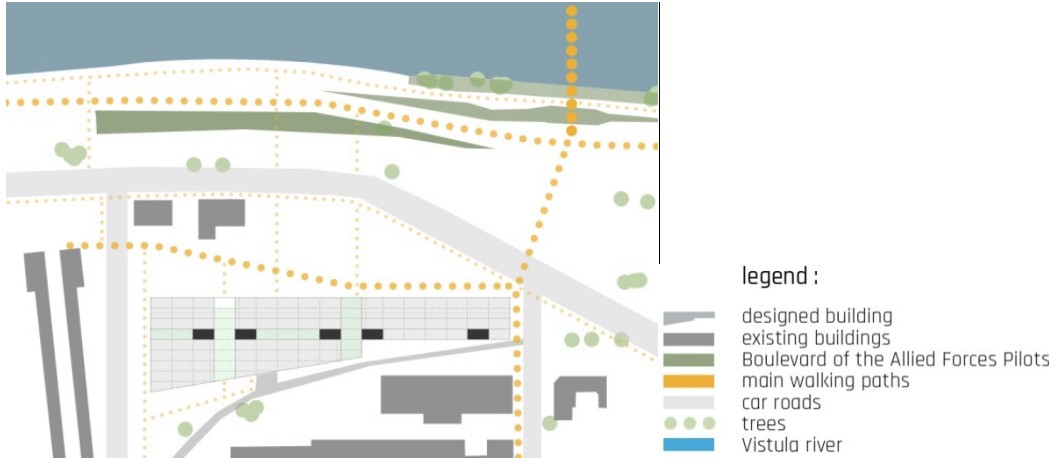

**Figure 21.** Diagrams showing the location of the study area–transport and circulation, (elaborated by E. Latusek).

*4.4. Programme*

The research conducted in this work contributed to the determination of needs and requirements within the scope of the following functions; museum, office, hotel and a local centre (Figure 22).

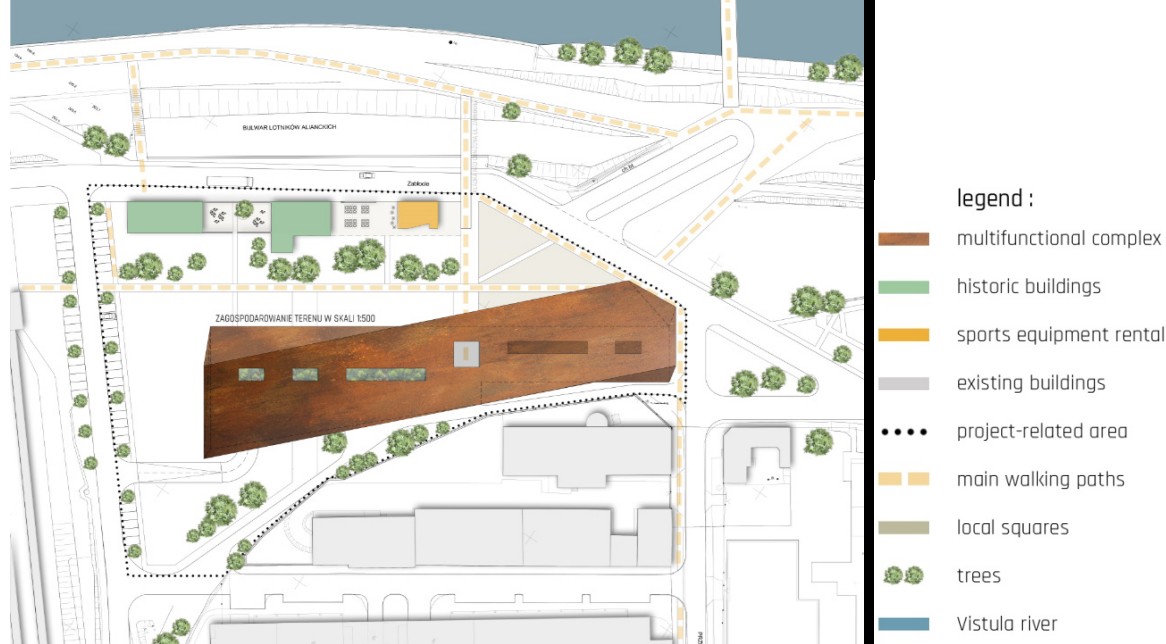

**Figure 22.** Designed land development, including the multifunctional complex and nearby objects, own elaboration (elaborated by E. Latusek).

Museum: Having conducted historical analysis, a decision was made to create a museum connected with the catastrophe of a Liberator aircraft and the history of the Allied Forces Pilots. This idea was based and preceded by activities that had been carried out since 1999, including the change of the name of the embankment's section into Boulevard of the Allied Forces Pilots (Bulwar Lotników Alianckich). The museum building may become a further unit of the Historical Museum of Cracow (Kraków) cooperating with other museums and cultural objects in the neighbourhood, such as MOCAK, Oskar Schindler's Enamel Factory, Cricoteka and Planeta Lem.

Offices: The conducted surveys of user groups indicated the necessity of the assumption that the newly designed offices are to feature functions of the space satisfying the needs of freelancers, as the

space of a coworking type, workplaces used for a certain amount of time and the space enhancing start-ups. Coworking makes it possible to work in a rented room or space. It is the space used most often by freelancers because it gives a bigger comfort of work than at home. Coworking centres can be already found in almost all large cities, also Cracow witnesses the appearance of a bigger and bigger number of such spaces. For instance, in the vicinity of the project-related area there are two such facilities: Studio Zabłocie 2 and Biznes Lab. The designed office part of the multifunctional complex provides for similar spaces, however, not competing with the existing ones.

Hotel: The formation of a local centre and planned development of the area in the scope of culture and promotion of local history will cause the influx of tourists, visitors, customers of the industrial plants and people using the office facilities. Due to this fact, the design provides for the creation of a four-star hotel in this area. Although the project Cracow Marina Marina Kraków provides for the hotel connected with sports and recreational functions, the function of the newly-designed hotel will be connected with office and conference purposes. Economic aspects of the introduced changes are as follows.

■　Multifunctional complex features functions satisfying the contemporary needs of users of this rapidly transforming district; this aims to attract young labour force and increase the attractiveness of this area in relation to other areas located in proximity.

■　Introduction of workplaces into the district which has a developing residential function will limit the traffic and circulation of inhabitants and thus positively influence the quality of life in the district.

■　Objects under conservation policy: restaurant (Zabłocie 13) as well as culture and community centre 'Warsztat' ('Workshop' Zabłocie 25) have been included in the development context as objects of historic value and constitute an added value to the developed space by attracting clients.

Social aspects of the introduced changes are as follows.

■　Representative character of the public space was highlighted as a local centre for the district inhabitants and visiting tourists.

■　Concept of the multifunctional complex enriches and revitalizes functions connected with culture, recreation and historical education. Being the "salon" of the local centre, this place and its well-adopted functions are expected to attract users and visitors with different needs and interests.

■　Multifunctional complex faces the Vistula (Wisła) river in order to create an attractive panorama of Zabłocie, which can be seen from the opposite riverbank; this constitutes an important element of the formation of spatial order.

■　Concept of land development takes into account the existing vistas on a local and urban scale.

■　Study area was developed with the preservation of its unique, on a European scale, character, in the form of a commonly accessible waterfront space called the Vistula Boulevards (Bulwary Wiślane).

*4.5. Spatial and Functional Solutions*

Thanks to its new building development, a post-industrial character of Zabłocie area is being transformed into a fashionable place for living and spending free time. The residents of the apartment buildings will also co-form and co-use this local centre. The figure below shows a diagram of the formation of the body of the designed multifunctional complex (Figure 23).

The ground floor of the designed complex houses rooms and facilities enabling community integration (Item a, Figure 24, orange colour), including activities such as: craftwork, organisation of presentations and training sessions, after school activities for children. The design provides for a large multifunctional room. The ground floor area also includes: two entrance zones to the office part (Item a, Figure 24, blue colour), the hotel part (Item a, Figure 24, green colour), an aperitif-bar and restaurant with cooking facilities, and the space of the museum exhibitions with the entrance zone (Item a, Figure 24, yellow colour).

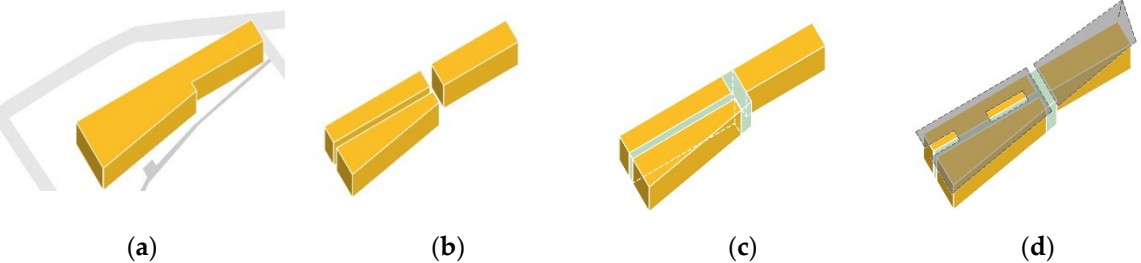

**Figure 23.** Diagram of the building body formation: (**a**) Adjustment of the object to the building plot after the delineation of new roads; (**b**) Division of the complex into museum, hotel and office functions; (**c**) Structural isolation of the 'wedge' with circulation and big groups of greenery; (**d**) Application of a flat roof and a spatial cover reminding of an aircraft wing (elaborated by E. Latusek).

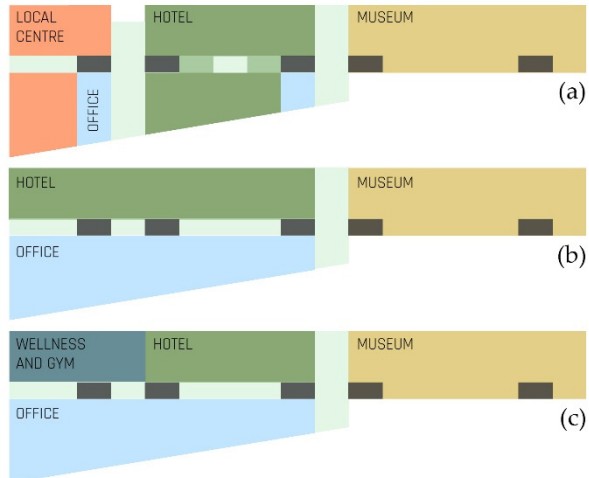

**Figure 24.** Functional diagram of the designed multifunctional complex: (**a**) ground floor; (**b**) repeatable storeys +1, +2; (**c**) storey +3 (elaborated by E. Latusek).

Storeys +1, +2 and +3 were designed in accordance with the division into three chief functions: office, hotel and museum. The last floor was supplemented with the space dedicated to biological renewal and bodybuilding to meet the requirements of the four-star hotel. The above-mentioned functions are connected by means of a structurally isolated "wedge" with big clusters of greenery and circulation. The roof reminds of an aircraft wing. It has openings providing natural lighting to the single-space "wedge" and the rooms on lower storeys. The main emphasis, however, was put on the proper design of the museum space. In its central part there is space for a large exhibit–the replica of a Liberator aircraft. There are also exhibition rooms connected with the local history.

*4.6. Comparative Analysis with the Existing European Examples of Development–Regeneration of Degraded Riverside Areas*

To undertake a discussion with the existing examples of riverside development, an analysis of several projects was attempted, such as Paris Rive Gauche, Oslo Fjord City & Akerselva River, Refshaleoen in Copenhagen. The above-mentioned areas considerably differ in their scale from the study area discussed herein; however, they introduce similar functional and spatial solutions.

### 4.6.1. Paris Rive Gauche

Paris Rive Gauche constitutes the new 13th municipal district of Paris located south of the river Seine. The area is an example of a high-quality urban-development space (including 10 hectares of green areas) and the integration with sustainable transport. The vicinity of Austerlitz Nord contributed to the construction of a series of office buildings with shops and catering establishments on their ground floors in order to meet needs of the local community for the creation of a local centre. Similarly in Kraków-Zabłocie, there is a railway station adjacent to the study area. The analyses conducted showed that there is also a need for the creation of an active local centre. Therefore, it was proposed that the coworking office centres should be built there (as a type of space most often used by freelancers), including a museum, a culture zone for the inhabitants and a hotel with conference and catering facilities.

Also, similarly to Cracow (Kraków), the development of Austerlitz Nord takes advantage of the difference in the terrain altitude (~9 m) using it for the passages for pedestrians from alleys to the waterfront. In the design of the Vistula Riverside development, a pedestrian and cycling footbridge (in a north–south direction) is planned over a heavy traffic artery located in a terrain depression. The footbridge is to facilitate traffic and circulation between the designed multifunctional centre and the riverside boulevard.

The National Library in the district Paris Rive Gauche is located in a similar urban-development context to the designed multifunctional complex in Kraków-Zabłocie. A massive body of the library is separated from the river by municipal infrastructure and a row of high trees. On the other hand, in Kraków-Zabłocie, due to the development of the terrain and introduction of changes in the road and pedestrian infrastructure, the safety of free circulation was improved between the designed building and vast green areas on the Vistula (Wisła) waterfront. Thanks to that, the space in front of the multifunctional complex freely blends with the riverside boulevard.

### 4.6.2. Oslo Fjord City & Akerselva River

Oslo Fjord City is one of the most interesting concepts of regeneration and creation of space of urban areas in the riverside part of Oslo downtown. Former harbour space was converted into the "salon" of the city with residential buildings interwoven with commercial objects. The area of Kraków-Zabłocie is undergoing similar transformations. Significant historic post-industrial sites were built with new residential and public utility objects resulting in the gradual increase in the number of users and inhabitants of this area in recent years.

The example of Oslo is important due to the way of merging public buildings with the waterfront. The central point is the Opera House—the largest building devoted to culture that has been built in Norway over the span of 700 years. This is a building whose roof was made available to users. It plays a function of a viewpoint and constitutes an integral part of the public space areas. The designed multifunctional complex in Cracow is supposed to play a similar role. Having conducted historical analyses, it was noticed that there was a need for the creation of an architectural dominant in the form of a multifunctional complex (including a museum, local centre, hotel and offices). The museum commemorating the catastrophe of a Liberator aircraft and the history of the Allied Forces Pilots is to serve the local community for the promotion of significant events connected with the history of Kraków-Zabłocie. Moreover, a multifunctional centre located in the area of the meeting point of three central districts of Cracow (Old Town, Grzegórzki, Podgórze) may be a magnet drawing new users to the southern district of Cracow (Kraków).

Another solution having similar features to the one applied in Kraków-Zabłocie is the development of the waterfront of the river Akerselva, which flows entirely within the boundaries of the city of Oslo. Upon this river, meandering through the municipal park areas, there is an object proving an interesting transformation of a degraded post-industrial silo into a students' residence hall. This exceptional building became the landmark of the city of Oslo and was awarded a prize in 2002. This shows how important it is for the space users to preserve characteristic features of the place. Similarly to the

Akerselva riverside, the Vistula (Wisła) riverfront witnesses architectural and urban planning changes. Historic housing development of the Podgórze district is being subjected to gradual regeneration: a museum was created in the former Oskar Schindler's Enamel Factory, the MOCAK museum was constructed, post-industrial objects were transformed into a students' residence hall 'Livinn Kraków' and an office centre, the area of the former railway station was transformed into a park 'Park Stacja Wisła'. The study area encompasses objects under conservation policy, i.e., a restaurant 'Zabłocie 13' and an independent culture and community centre 'Warsztat' (Workshop), which are included in the context of the new development as representing historic and social values.

### 4.6.3. Refshaleoen in Copenhagen

Refshaleoen is a flourishing place where its users may spend leisure time in an attractive way. Once degraded, the warehouses and space remaining after historic shipyard Burmeister & Wain are nowadays filled with private business which has brought in fresh commercial energy and serves the local community. This new fashionable district of Copenhagen is located just 15 min away by bicycle from the city centre. The example of the thriving island of Refshaleoen confirms that areas located in some distance from the city centre may be sufficiently attractive to appeal to many users. A similar distance must be covered to reach the area of Zabłocie from the centre of Cracow. The proposed concept of the Vistula riverside development aims to introduce solutions which will regenerate the degraded areas in terms of economic, social and pro-ecological issues.

## 5. Conclusions

In many European cities there is a noticeable trend to build compact multifunctional complexes and public buildings in degraded areas. Not only does this help to save costs and urban space, but also enables mixing of functions and the community of users. As it turns out, local urban centres may be created not only in open public space but also in the form of 'city salons' within the structure of objects. Multifunctionality makes it possible to mix different groups of users having various needs and interests, as for instance in the Lucerne Culture and Convention Centre KKL or the Oslo Opera House.

Attractiveness of the regenerated spaces is increased by objects having characteristic appearance. Similarly to the case of converting a silo into a students' residence hall in Oslo, in the study area in Kraków-Zabłocie, the buildings under conservation policy (Restaurant 'Zabłocie 13' as well as Culture and Community Centre 'Warsztat'–'Workshop') were included as historically valuable objects in the context of the development.

The concept of the development of Podolski Boulevard (Bulwar Podolski) in Kraków-Zabłocie and turning it into a multifunctional complex meets the needs of the local community and enables further development of tourism, which is so important to Cracow. The analysis of the diversified design and social issues related to the study area aimed at the creation of an interesting programme as well as attractive functional and spatial solutions. In addition, historic events connected with the study area made it possible to assign a special character to this place. The concept of the development of the Vistula (Wisła) riverside including the design of a multifunctional complex having pro-social, museum, hotel and coworking functions resulted from the conducted holistic research. This made it possible to achieve expected results in compliance with vital urban planning documents for this area and the idea of sustainable development in terms of ecological, economic and social aspects.

**Author Contributions:** Conceptualization, B.M.-P.; Data curation, E.L. and B.M.-P.; Formal analysis, B.M.-P.; Funding acquisition, E.L. and B.M.-P.; Investigation, E.L.; Methodology, E.L.; Project administration, E.L.; Resources, E.L.; Software, E.L.; Supervision, B.M.-P.; Validation, B.M.-P.; Visualization, E.L.; Writing—original draft, E.L.; Writing—review & editing, B.M.-P. All authors have read and agreed to the published version of the manuscript.

**Funding:** This research received no external funding.

**Conflicts of Interest:** The authors declare no conflicts of interest.

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
