# Peer review of "A Concept of the Development of Riverside Embankment in the Context of Cracow (A Local Centre)"

_buildings, doi:10.3390/buildings10030056_

Round 1

Reviewer 1 Report

The proposed article (as a practice-oriented paper) has interesting and rather original content.

Anyway, the author should devote more attention to research and methods rather than project descriptions.

The author should explain how his ‘mixed own method’ (line 17 and Table 1) differs from others and what it brings to the discussion on riverfront regeneration. A comparison with the countless riverside projects (at least European ones) could be useful to highlight the characteristics and results of the author's case study. Examples like: Paris Rive Gauche’; Oslo Fjord City; Tagus Estuary riverside regeneration (Lisbon Metropolitan Area); Rotterdam Old Ports; IJ Waterfront of North Amsterdam, to name a few, could support a productive discussion.

_Keywords should be more specific, I suggest joining words into sentences (for example:Krakow; Zabłocie vs. Krakow Zabłocie).

_All captions should mention the sources (e.g. author’s drawing...)

_Subsection 2.6. appears two times.

Author Response

Response to Reviewer 1 Comments

Point 1: The proposed article (as a practice-oriented paper) has interesting and rather original content.

Response 1: ---

Point 2: Anyway, the author should devote more attention to research and methods rather than project descriptions.

Response 2: The research and methods were supplemented with additional information in accordance with the Reviewer’s guidelines.

Line 351 , the following text was introduced: “3. Conclusions based on investigations

Line 363 , the following text was introduced:

“3.1. Guidelines

In the Study of Conditions and Directions of Spatial Development of the City of Cracow (Kraków),  the analysed area is located within the boundaries of two structural urban-planning units. The delineated roads are to correspond to the guidelines provided in the local development plans. The above-mentioned guidelines for the local development plans define also the category of the analysed area as the services areas. Their primary purpose is to become the building development intended for the following functions: offices, culture and other services along with some necessary ancillary buildings and accompanying greenery. The optional function includes arranged or unarranged green spaces, such as parks, squares, greenery spots, river parks as well as green belts around buildings and vegetation screens (the so called ‘vegetation barriers’.)

The Resolution of the Council of the City of Cracow (Kraków) of 28 June 2006 on Passing the Local Plan of Spatial Development of the Zabłocie Area – general regulations constitute that the existing building development should be preserved or rebuilt, new building development should be implemented and new investments made, the changes in the use and development of the land should be introduced. The existing valuable buildings and areas can be utilised in the previous way, until these areas are newly developed according to plan. The whole area covered by the plan is located on the terrain where there is a risk of landslide. No admissible noise levels in the environment were defined for this area in the development plan. The space of the site needs putting in order, that is the integration of land plots and removal of some derelict buildings. The objects under conservation, namely a restaurant (Zabłocie 13) and a culture and community centre ‘Workshop’ (‘Warsztat’) (Zabłocie 25) may be included in the development context as  objects of historical significance.

[PICTURE]

Figure 17. Plots constituting the main part of the development of a multi-functional complex. The area includes objects and sites under conservation (elaborated by E. Latusek).

Point 3: The author should explain how his ‘mixed own method’ (line 17 and Table 1) differs from others and what it brings to the discussion on riverfront regeneration.

Response 3:

Line 86 , the following text was introduced:

Table 1. List of types of activities within the method of logical argumentation and heuristic methods, (table by E. Latusek, on the basis of [1]).

Type of activity

Method of logical argumentation

Heuristic methods

Research issues

Theoretical and analytical studies, theory development, humanistic and philosophical interpretation of architectural issues, theory of architecture.

Diagnosis of the present state and prediction of directions of development, search for key developmental factors, development of scenarios.

Activities

undertaken

Use of logic, analysis, deduction, synthesis, analogies, drawing conclusions, logical abstraction, optimization, logical abstract analysis.

Examination of expert opinions.

Research approach

Search for theoretical interpretation of objective or abstract facts. Researcher does not enter into any social interactions.

Sharing one’s knowledge and opinions.

Techniques applied

Description, explanation, logical interpretation, comparative studies,

grade scales.

Workshops, SWOT analysis, scenarios, ‘brainstorm’ meetings,  Delphi technique, foresight.

Tools applied

Subject literature, architectural and urban-planning documentation, computer and software programmes, comparisons, tables.

Surveys, recorders, Internet.

Expected results

Presentation of logical conclusions (academic approach). Approaches of design optimization; mathematical theories; algorithms. Description of the problem and its interpretation, development of procedure algorithm, publication.

Charting possible directions of development (empirical approach). Determination of implementation methods of strategic goals (normative approach).

Line 95 , the following part was changed:

“As a result, there appeared a concept of creating own mixed research method , which is based on a fragmentary division of two kinds of activities described above.”

Line 98 the following part was changed: “Table 2. Development of own mixed method on the basis of heuristic methods and the method of logical argumentation, (table by E. Latusek).”

Point 4: A comparison with the countless riverside projects (at least European ones) could be useful to highlight the characteristics and results of the author's case study. Examples like: Paris Rive Gauche’; Oslo Fjord City; Tagus Estuary riverside regeneration (Lisbon Metropolitan Area); Rotterdam Old Ports; IJ Waterfront of North Amsterdam, to name a few, could support a productive discussion.

Response 4:

Line 364 the following part was changed: “4. Discussion – proposed solutions “

Line 30 the following text was introduced: “Comparative studies were conducted including selected examples of European riverside development projects.”

Line 377 the following text was introduced:

“In order to generate a discussion within the scope of the existing examples of the development of riverside areas, the analysis of the following projects was attempted: Paris Rive Gauche, Oslo Fjord City & Akerselva river, Refshaleoen in Copenhagen.

Paris Rive Gauche

Paris Rive Gauche constitutes the new 13th municipal district of Paris located south of the river Seine. The area is an example of a high-quality urban-development space (including 10 hectares of green areas) and the integration with sustainable transport. Within the boundaries of the district, on the river bank, there are spectacular public buildings, such as the French National Library. The library is located in an urban-development context which is similar to that of the newly designed multi-functional complex in the city of Cracow (Kraków). The latter, however, is not separated from the river by urban infrastructure but it freely blends into the riverside embankment (boulevard) through the land development and landscaping.

Oslo Fjord City & Akerselva river

Oslo features an interesting phenomenon of space creation, namely the project of the regeneration of urban areas located on the downtown riverside - Oslo Fjord City. Former harbour space was converted into the ‘salon’ of the city with residential buildings interwoven with commercial objects. This example is significant due to the way of linking public buildings with the riverside areas. The focal point is the Opera House being the largest cultural object which has been built in Norway in 700 years. The building’s roof has been made accessible to users and plays a function of a viewpoint being at the same time an integral part of the public space.

[PICTURE]

Figure 19. (a) Opera House in Oslo, (b) Oslo Fjord City, (photo by B. Majerska).

The river Akerselva featuring many cascades and waterfalls flows entirely within the city boundaries. Upon this river meandering through the municipal park areas there is an object proving an interesting transformation of a degraded post-industrial silo into a students’ residential hall.

[PICTURE]

Figure 20. Students’ Residence Hall on the river Akerselva (photo by E. Latusek).

Refshaleoen in Copenhagen

In 1871, the shipyard of Burmeister & Wain was founded on the island of Refshaleoen. In its peak period it employed several thousand workers. The former icon of the Danish industrial history was closed down in the 1990s, which resulted in the abandonment of many warehouses and factory buildings. Presently, Refshaleoen is a thriving place where its users can spend their free time in an attractive way. There are riverside restaurants with their own piers where the inhabitants can lounge on sunny days. Once degraded warehouses and the space between them are now filled with private businesses which have brought in fresh commercial energy and serve the local community.”

[PICTURE]

Figure 21. Sluseholmen street in Copenhagen, (photo by B. Majerska).

Point 5: Keywords should be more specific, I suggest joining words into sentences (for example: Krakow; Zabłocie vs. Krakow Zabłocie).

Response 5:

Line 37 the following part was changed: “Kraków Zabłocie; Podolski Boulevard; development of riverside embankment; downtown riverside areas; urban local centre; community; historical context; multifunctional complex.”

Point 6: All captions should mention the sources (e.g. author’s drawing...)

Response 6:

Figure 1. Diagrams showing the location of the project-related area: (a) Podgórze district area in the context of other Cracow districts; (b) Zabłocie area within the scope of Podgórze district and in proximity of the Old Town [], (elaborated by E. Latusek).

Figure 2. Fragment of the regeneration area, sub-area of Stare Podgórze – Zabłocie, own elaboration E. Latusek on the basis of the map extract from the Study, (elaborated by E. Latusek).

Figure 3. Objects recorded in the heritage register, (photo by E. Latusek).

Figure 4. Potentially important places in the vicinity of the project-related area: (1) Project-related area, (2) Boulevard of the Allied Forces Pilots (Bulwar Lotników Alianckich); (3) Planned memorial of the Allied Forces Pilots; (4) The project-related area borders with a lighting company; (5) New residential quarters ATAL Residence and Garden Residence; (6) Park ‘Vistula Station’ (‘Park Stacja Wisła’) which gained an award in the contest for the best developed space in Poland; (7) Planned Cracow Marina (‘Marina Kraków’); (8) Planned flyover for pedestrians and bicycles; (9) Railway station Kraków Zabłocie; (10) Kotlarski Bridge (Most Kotlarski), (elaborated by E. Latusek).

Figure 5. Boulevard of the Allied Forces Pilots (Bulwar Lotników Alianckich) within Podolski Boulevard (Bulwar Podolski) only partially covered by the Local Plan of Spatial Development of the area called ‘Vistula Boulevards’ (‘Bulwary Wisły’) [], (elaborated by E. Latusek).

Figure 6. Main arterial roads of Cracow (Kraków), (elaborated by E. Latusek).

Figure 7. Railways in Cracow (Kraków), (elaborated by E. Latusek).

Figure 8. Significant characteristic building development surrounding the study area, (elaborated by E. Latusek).

Figure 9. Main vistas in the context of Podolski Boulevard (Bulwar Podolski), (elaborated by E. Latusek).

Figure 10. Plots constituting the main part of the development of the design of a multi-functional complex with objects and sites under the conservation, (photo by E. Latusek).

Figure 11. Cultural dominants of Podolski Boulevard (Bulwar Podolski): (1) Project-related area; (2) Oskar Schindler’s Enamel Factory; (3) Museum of Contemporary Art in Kraków - MOCAK; (4) Centre for the Documentation of the Art of Tadeusz Kantor – Cricoteka; (5) Centre of Literature and Language – ‘Planet Lem’ (‘Planeta Lem’); (6) Zabłocie Business Park; (7) Student Hall of Residence Livinn Cracow; (8) Park ‘Vistula Station’ (‘Stacja Wisła’); (9) ATAL Residence; (10) Cracow Marina, (elaborated by E. Latusek).

Figure 12. Museum - Oskar Schindler’s Enamel Factory in Cracow (Item 10, Fig. 1), (photo by E. Latusek).

Figure13. Museum of Contemporary Art in Kraków - MOCAK, designed by Claudio Nardi Architette (Item 10, Fig. 1), (photo by E. Latusek).

Figure 14. Centre of the Documentation of the Art of Tadeusz Kantor - Cricoteka, designed by Biuro Architektoniczne Wizja and nsMoonStudio (Item 10, Fig. 1), (photo by E. Latusek).

Figure 15. Student Residence Hall Livinn Kraków, renovation and modernisation by Unibep (Item 10, Fig. 1), (photo by E. Latusek).

Figure 16. Park Vistula Station (Park Stacja Wisła), designed by Michał Grzybowski. In the background, residential building ATAL Residence Kraków, designed by Biuro Rozwoju Krakowa S.A. (Item 10, Fig. 1), (photo by E. Latusek).

Figure 17. Plots constituting the main part of the development of the concept of a multi-functional complex with objects and sites under the conservation, (elaborated by E. Latusek).

Figure 18. Design concept in the context of surrounding building development: (1) Project-related area; (2) Crikoteka; (2) Qubus Hotel; (3) Cracow Academy, (elaborated by E. Latusek).

Figure 19. (a) Opera House in Oslo, (b) Oslo Fjord City, (photo by B. Majerska).

Figure 20. Photograph:), (b) Students’ Residence Hall on the river Akerselva (photo by E. Latusek).

Figure 21. Sluseholmen street in Copenhagen, (photo by B. Majerska).

Figure 22. Analysis of greenery, (elaborated by E. Latusek).

Figure 23.  Embedding the object in the context: (1) Flood risk area, (2) Designed multifunctional complex, (3) Designed sports equipment rental, (4) Existing historic buildings, (5) Existing objects of the lighting company, (elaborated by E. Latusek).

Figure 24. Diagrams showing the location of the study area – transport and circulation, (elaborated by E. Latusek).

Figure 25. Designed land development, including the multifunctional complex and nearby objects, own elaboration, (elaborated by E. Latusek).

Figure 26. Diagram of the building body formation: (a) Adjustment of the object to the building plot after the delineation of new roads; (b) Division of the complex into museum, hotel and office functions; (c) Structural isolation of the ‘wedge’ with circulation and big groups of greenery; (d) Application of a flat roof and a spatial cover reminding of an aircraft wing, (elaborated by E. Latusek).

Figure 27. Functional diagram of the designed multifunctional complex: (a) ground floor; (b) repeatable storeys +1,+2; (c) storey +3, (elaborated by E. Latusek).

Table 1. List of types of activities of the method of logical argumentation and heuristic methods, (table by  E. Latusek, based on [[2]]).

Table 2. Development of own mixed method on the basis of heuristic methods and the method of logical argumentation, (table by E. Latusek).

Table 3. Grupy użytkowników obszaru opracowania i ich potrzeby, (table by  E. Latusek).

Table 4. Analiza SWOT dotycząca stanu istniejącego obszaru objętego opracowaniem, (table by  E. Latusek).

Point 7: Subsection 2.6. appears twice.

Response 7: The sequence of the sub-sections was changed according to the Reviewer’s instructions:

Introduction Materials and methods

2.1.                  Research questions

2.2.                  Historical research

2.3.                  Examination of the contemporary state

2.4.                  Studies of user groups

2.5.                  Research on the local community’s initiatives

2.6.                  Legal and formal status

2.7.                  Research on urban-development  composition

2.8.                  Urban-development dominants

2.9.                  Major vistas

2.10.                Research on greenery elements

2.11.                Research on architecture

Research results (the point was deleted)

2.12.                SWOT Analysis

Conclusions based on investigations

3.1.                  Guidelines

Discussion and proposed solutions

4.1.                  Greenery

4.2.                  Cubature buildings

4.3.                  Transport and circulation

4.4.                  Programme

4.5.                  Spatial and functional solutions

Summary

[1] Tymkiewicz J. Metody badań w architekturze - Methods of Research in Architecture; Lecture in the subject: Methodology of Research Work, Faculty of Architecture at the Silesian University of Technology in Gliwice - Wydział Architektury Politechniki Śląskiej, Gliwice, Poland, 23.02.2018

[2] Tymkiewicz J. Metody badań w architekturze - Methods of Research in Architecture; Lecture in the subject: Methodology of Research Work, Faculty of Architecture at the Silesian University of Technology in Gliwice - Wydział Architektury Politechniki Śląskiej, Gliwice, Polska, 23.02.2018

Reviewer 2 Report

The article presents the concept of development of degraded riverside areas in the context of undertaking rational design activities. The discussions are conducted on the basis of multi-aspect analyses and author's research. The article is suitable for publication, however, in the opinion of the reviewer, it requires consideration of several amendments to make the research problem more readable.
1. The introduction requires clear articulation of the subject of the work and the spatial scope of the area covered by the research.
2. The research and analytical part of the paper should be supplemented with elements of greenery occurring in the area of the study and elements of urban composition determining the adopted solutions.
3. The summary, apart from the general summary, should contain detailed conclusions resulting from the comparative analysis of the presented research.
4. Figures 1, 3, 4, 5, 11, 13, 14 included in the article require legends supplementing the presented graphic material.
5. Table no. 3 - in the area of strengths and weaknesses - point 3 is written down identically - this requires a commentary.

Author Response

Response to Reviewer 2 Comments

Point 1: The article presents the concept of development of degraded riverside areas in the context of undertaking rational design activities. The discussions are conducted on the basis of multi-aspect analyses and author's research. The article is suitable for publication, however, in the opinion of the reviewer, it requires consideration of several amendments to make the research problem more readable. 

Response 1: The text was edited in accordance with the Reviewer’s recommendations.

Point 2: The introduction requires clear articulation of the subject of the work and the spatial scope of the area covered by the research.

Response 2:

Line 56 the following text was introduced:

“The subject of the work is a fragment of the riverside embankment of the river Vistula (Wisła) in Cracow (Kraków). This riverfront is in many respects a very interesting area, which has not been properly developed yet. This particular site was selected due to its purpose: in urban-planning documents this place is intended for the function of a local centre. It is a socially significant place. This requires a proper approach to designing with a view to the creation of architecture. After the research and spatial analyses, it was decided that a multi-functional complex should be designed to meet the needs of Cracow’s inhabitants and visiting tourists. This work and investigations concern the area which  was designated as a local centre in the Study of Conditions and Directions of Spatial Development of the City of Cracow (Kraków).

[PICTURE]

Figure 2. Fragment of the regeneration area, sub-area Stare Podgórze – Zabłocie, own elaboration by E. Latusek on the basis of the map extract from the Study, (elaborated by E. Latusek), (elaboration by E. Latusek).

Point 3: The research and analytical part of the paper should be supplemented with elements of greenery occurring in the area of the study and elements of urban composition determining the adopted solutions.

Response 3:

Line 265 the following text was introduced:

"2.7. Research on urban-development  composition

Both national and European roads run through Cracow (Kraków). Typical traffic intensity during rush hours does not exceed critical limits. High traffic intensity occurs along the second ring road of Cracow, in Gustawa Herlinga-Grudzińskiego street and in the vicinity of the Kotlarski Bridge and the Marshal Józef Piłsudski Bridge. However, traffic jams appear on the regional road no. 776 in Powstańców Wielkopolskich street.

[PICTURE]

Figure 6. Main arterial roads of Cracow (Kraków), (photo by E. Latusek).

Cracow (Kraków) is one of the largest railway interchange stations in Poland. It is linked to the majority of cities in Poland, including express Pendolino links with Warsaw (Warszawa) and Gdańsk. In addition, it has international connections with Vienna, Prague, Budapest and Lviv. The Main Railway Station in Cracow along with the Małopolska Region Coach Station, municipal public transport (buses, underground fast tram) and the link to the Cracow-Balice International Airport make up a complex called the Cracow Public Transport Centre. By the end of 2020, four new rail tracks will have been built on two newly-constructed railway trestle bridges on the crosstown line. The Polish Railways PKP Polskie Linie Kolejowe S.A. (Joint Stock Company) link the central railway station with the station Kraków-Płaszów facilitating thus the traffic of agglomeration and long-distance trains. The station Kraków-Zabłocie is currently under modernization, which is connected with the above-mentioned investment.

[PICTURE]

Figure 7. Railways in Cracow (Kraków), (photo by E. Latusek).

Poland is to ultimately house five green bicycle trails referred to as – greenways. Local bicycle loops have been opened on the Amber Greenway (Szlak Bursztynowy: Budapest - Bańska Szczawnica - Cracow - Gdańsk). There is an international trail Cracow-Moravia-Vienna, being an eco-touristic corridor exhibiting the cultural, natural and historic heritage of Central Europe. In the future, the aforesaid corridor should become the longest ‘alley of trees’ in Europe. In the direct vicinity of the study area there are many local bicycle lanes and a public bike rental system.

Along the river Vistula (Wisła) in Kraków there is an inland shipping route known as ‘Waterway of the Upper Vistula River’ (Droga Wodna Górnej Wisły). In 2018, the Ministry of Marine Economy and Inland Navigation signed an agreement for the development of a transport analysis, being the first study of this type in relation to inland water transport. The analysis should concern inland navigation on the river Odra and Wisła, as another element of the programme aimed to develop inland waterways in Poland. The city of Cracow, encouraged by its residents, is planning to implement a project named ‘Marina Kraków’.

2.8. Urban-development dominants

Near the study area there are three new high-standard residential complexes. South of the area there are spaces with strong historical connotations: Cricoteka, Ghetto Heroes Square, a concept to create the Planet Lem object, Oscar Schindler’s Enamel Factory, Museum of Contemporary Art in Cracow - MOCAK. Nearby large educational establishments include the Andrzej Frycz Modrzewski Higher School in Cracow, the Institute of Ceramics and Building Materials, Glass and Building Materials Division in Kraków-Podgórze, AMA Film Academy, students’ dormitory of the Academy of Music in Cracow and the Adam Mickiewicz Secondary School of General Education no. 4. Nearby hotels include 4-star standard Qubus Hotel, Hotel Galaxy, PURO Hotel Kraków Kazimierz, INX Design Hotel as well as many other hotels located in the district of Kazimierz. Nearby recreational facilities  include shopping mall Galeria Kazimierz, Saturn Fitness, Gym Park fitness centre, FitNOW fitness centre and dietician’s, Laserpark - laser entertainment centre and, located by the river Wisła: a water tram stop and a kayak rental point. On the opposite bank of the river Wisła there is Galeria Kazimierz shopping mall, which, in the future, will be accessible via a footbridge (for pedestrians and cyclists).

[PICTURE]

Figure 8. Significant characteristic building development surrounding the study area, (photo by E. Latusek).

2.9. Major vistas

The area of Podolski Boulevard (Bulwar Podolski) is located between the mouth of the Wilga river and the railway bridge in Zabłocie in the district of Podgórze. The Local Development Plan for the Area of the Vistula Riverside, the so-called ‘Wisła Boulevards’, contains regulations related to land development, yet only in relation to the western part of Podolski Boulevard, without its eastern part located in Zabłocie. This part of riverside including areas located east of the railway bridge has not been regulated in terms of land development, supplementation of landscape architecture and lighting, adjustment of greenery. The general plan provides for related supplementation as well as the maintaining of main passageways and viewpoints.

[PICTURE]

Figure 9. Main vistas in the context of Podolski Boulevard (Bulwar Podolski), (photo by E. Latusek).

2.10. Research on greenery elements

The preliminary analyses related to the study area provoked a number of questions, one of which is concerned with the lack of the appropriate development plan for Podolski Boulevard. The detailed assessment of the existing condition revealed that an intended space of historical commemoration was to be the Boulevard of Allied Forces Pilots (Bulwar Lotników Alianckich). Today, this area is still wasteland (overgrown with grass and high greenery) despite the fact that nearby there are new apartment buildings and the Andrzej Frycz Modrzewski Higher School in Cracow. Only a plaque with the boulevard name stresses the significance of the area. Because of the fact that Zabłocie is characterised by the high-density housing development of the city centre and yet does not have a payable parking zone, today the boulevard is often used as a ‘wild’ car park.

[PICTURE]

Figure 10. Plots constituting the main part of the development of the design of a multi-functional complex with objects and sites under the conservation, (photo by E. Latusek).

Line 265 the following change was made: “2.11. Architectural research

Line 343 the following text was deleted: “3. Research results

Line 344 the following change was made: “2.12. SWOT Analysis

Line 377 the following text was introduced:

“4.1.  Greenery

The development of the document ‘Trends of the Development and the Management of Green Areas in Kraków in the Years 2017-2030’ involved the development of the electronic space-related ‘Concept of the Public Green Areas System’ and ‘Register of Green Areas’, helpful in the continuous management and maintenance of the areas of greenery by the Department for Municipal Greenery Management. As can be seen in the diagram below, a large concentration of greenery is located in the Park ‘The Vistula Station’ (Park Stacja Wisła). Along the Podolski Boulevard there are single groups of trees. Only the area designated for the ‘Marina Kraków’ investment is characterised by the greater density of riverside greenery and high trees. The study area is intended to be planted with many trees and shrubs. The study area will be maintained as the area of ecological and landscape greenery.

[PICTURE]

Figure 22. Analysis of greenery, (photo by  E. Latusek).

Line 407 the following change was made:

[PICTURE]

Figure 24. Diagrams showing the location of the study area – transport and circulation, (photo by E. Latusek).

Point 4: The summary, apart from the general summary, should contain detailed conclusions resulting from the comparative analysis of the presented research.

Response 4:

Line 468 the following change was made: “5. Summary

Line 469 the following text was introduced:

“In many European cities, it is possible to notice a tendency to build multi-purpose complexes and public buildings. Such an approach not only makes it possible to reduce costs and urban space but also enables the mixing of functions and users’ community. As it turns out, local urban centres can be created not only in public space but also in buildings themselves. Multifunctionality enables the mixing of various groups of users having multiple interest as, e.g. at the Lucerne Culture and Congress Centre (KKL) or the Oslo Opera House.

The attractiveness of regenerated spaces is often enhanced by characteristic buildings. Similar to the conversion of a silo into a students’ dormitory in Oslo, it is assumed that the study area will contain two buildings under conservation area policy, i.e. the restaurant “Zabłocie 13” and the culture and community centre “Warsztat”, included in the context of the land development plan as representing historic values.

The bustling Refshaleoen island is just a 15-minute bike ride away from the city centre of Copenhagen. A similar distance is between Zabłocie and the centre of  Kraków. As can be seen, areas located slightly away from the city centre can still attract many users.

The National Library in the district of Paris Rive Gauche is situated in a similar urban context as the designed multi-purpose complex in Zabłocie in Kraków. The enormous building of the National Library is separated from the river by the city infrastructure and an espalier of high trees. In the developed concept, through the land development which introduces changes in the road and pedestrian infrastructure as well as by facilitating circulation between the building and the areas of greenery, the space in front of the multi-functional complex uninterruptedly joins the riverside boulevard.

Point 5: Figures 1, 3, 4, 5, 11, 13, 14 included in the article require legends supplementing the presented graphic material.

Response 5:

Figure 1

[PICTURE]

Figure 1. Schematy obrazujące położenie obszaru objętego opracowaniem: (a) Obszar Podgórza w kontekście dzielnic Krakowa; (b) Obszar Zabłocia w obrębie dzielnicy Podgórze i bliskości Starego Miasta [[1]],(fot. E. Latusek).

Figure 3

[PICTURE]

Figure 3 Figure 4. Miejsca potencjalnie ważne w pobliżu obszaru opracowania: (1) Obszar opracowania, (2) Bulwar Lotników Alianckich; (3) Planowany Pomnik Lotników Alianckich; (4) Teren opracowania graniczy z firmą branży oświetleniowej; (5) Nowe kwartały mieszkaniowe ATAL Residence i Garden Residence; (6) Park Stacja Wisła nagrodzony w konkursie na najlepiej zagospodarowaną przestrzeń w Polsce; (7) Planowana Marina Kraków; (8) Planowana Kładka pieszo-rowerowa; (9) Stacja kolejowa Kraków Zabłocie; (10) Most Kotlarski, (oprac. E. Latusek).

Figure 4

[PICTURE]

Figure 4 Figure 5. Bulwar Lotników Alianckich w obrębie Bulwaru Podolskiego tylko częściowo objętego Miejscowym Planem Zagospodarowania Przestrzennego obszaru „Bulwary Wisły” [[1]],(fot. E. Latusek).

Figure 5

[PICTURE]

Figure 5 Figure 11. Dominanty kulturalne Bulwaru Podolskiego: (1) Obszar objęty opracowaniem; (2) Fabryka Emalia Oskara Schindlera; (3) Muzeum Sztuki Współczesnej w Krakowie; (4) Ośrodek Dokumentacji Sztuki Tadeusza Kantora – Cricoteka; (5) Centrum Literatury i Języka – Planeta Lem; (6) Zabłocie Business Park; (7) Akademik Livinn Kraków; (8) Park Stacja Wisła; (9) ATAL Residence; (10) Marina Kraków, (fot. E. Latusek).

Figure 11

[PICTURE]

Figure 11 Figure 18. Koncepcja projektowa w kontekście otaczającej zabudowy: (1) Obszar objęty opracowaniem; (2) Crikoteka; (2) Qubus Hotel; (3) Akademia Krakowska, (fot. E. Latusek).

Figure 13

[PICTURE]

Figure 13 Figure 24. Schematy obrazujące położenie obszaru objętego opracowaniem - komunikacja piesza i samochodowa, (fot. E. Latusek).

Figure 14

[PICTURE]

Figure 14 Figure 25. Projektowane zagospodarowanie terenu z kompleksem wielofunkcyjnym i obiektami towarzyszącymi, opracowanie własne, (fot. E. Latusek).

Point 6: Table no. 3 - in the area of strengths and weaknesses - point 3 is written down identically - this requires a commentary.

Response 6:

Line 350 the following change was made in the strengths: “3. Diversified riverside boulevard topography (vertical retaining wall, gently inclined embankment) – as attractive elements of the development.”

Line 350 the following change was made in the weaknesses:  “3. Diversified riverside boulevard topography (vertical retaining wall, gently inclined embankment) – as a design-related impediment and a cost-increasing factor.”

Round 2

Reviewer 1 Report

The author should devote more attention to comparisons with other riverside projects (to highlight the characteristics and results of the author's case study) rather than project descriptions.

Taking into account that section 5 is closing the article, the author should use another title (original title for section 5: ‘Summary’

Author Response

Point 1: The author should devote more attention to comparisons with other riverside projects (to highlight the characteristics and results of the author's case study) rather than project descriptions.

Response 1:

Line 501-537 the text was deleted.

Line 640 the following text was introduced:

4.6. Comparison with the existing European examples

In order to generate a discussion within the scope of the existing examples of the development of riverside areas, the analysis of the following projects was attempted: Paris Rive Gauche, Oslo Fjord City & Akerselva river, Refshaleoen in Copenhagen. The study area considerably differs in its scale from other presented examples, however, the area development encompasses similar types of solutions.

Paris Rive Gauche

Paris Rive Gauche constitutes the new 13th municipal district of Paris located south of the river Seine. The area is an example of a high-quality urban-development space (including 10 hectares of green areas) and the integration with sustainable transport. One of the parts of this multi-aspect project is Austerlitz Nord located in close vicinity of the Austerlitz railway station. In both cases, i.e. Paris Rive Gauche - Austerlitz Nord and Zabłocie, the proximity of railway stations had influence on the implementation of functions of services and offices. This resulted in the construction of office buildings with shops and restaurants located on their ground floors. The development of Austerlitz Nord takes advantage of the difference in the terrain altitude (approx. 9 m) using it to build passages for pedestrians from alleys to the waterfront. The project of the Vistula Riverside also takes into account the diversity of landscape. In recent years, due to larger density of housing, there has been a noticeable increase in traffic on the  local road of the area (Zabłocie street). The design proposes to lower a fragment of this road in order to to build a flyover above it (in a north-south direction). The flyover will serve as a crossing for pedestrians and bicycles, which will facilitate the circulation between the designed building and the riverside boulevard.

Oslo Fjord City & Akerselva river

Oslo features an interesting phenomenon of space creation, namely the project of the regeneration of urban areas located on the downtown riverside - Oslo Fjord City. Former harbour space was converted into the ‘salon’ of the city with residential buildings interwoven with commercial objects. The area of Zabłocie in Cracow (Kraków) is undergoing similar transformations. Historically significant, post-industrial areas have been developed by building new residential objects, which has led to a steady increase in the number of the area users over the past few years.

This example is significant due to the way of linking public buildings with the riverside areas. The central point is the Opera House - the largest building devoted to culture which has been built in Norway over the span of 700 years. The building’s roof has been made accessible to users and plays a function of a viewpoint being at the same time an integral part of the public space. The designed multi-functional complex located on the Vistula (Wisła) riverfront is to play a similar role. Having conducted historical analyses, it was decided to introduce a function which will serve the local community and promote significant events connected with the history of Zabłocie. Moreover, this area borders with three central districts of Cracow (Old Town, Grzegóżki, Podgórze). Thanks to that, the multi-functional complex (consisting of a museum, local centre, hotel, offices) may become a magnet attracting new users to the southern district of Kraków.

[PICTURE]

The river Akerselva featuring many cascades and waterfalls flows entirely within the city boundaries. Upon this river meandering through the municipal park areas there is an object proving an interesting transformation of a degraded post-industrial silo into a students’ residential hall. This exceptional building became an icon and was awarded an architectural prize of the city of Oslo in 2002. This shows how important it is for the space users to preserve characteristic features of the place. Similarly to the Akerselva riverside, the Vistula (Wisłą) riverfront witnesses architectural and urban-planning changes. Historic housing development of the Podgórze district is being subjected to gradual revitalization. New iconic buildings come into being, such as: a museum in the former Oskar Schindler’s Enamel Factory, the MOCAK museum, post-industrial objects transformed into a students’ residence hall ‘Livinn Kraków’ and an office centre, the area of the former railway station transformed into a park ‘Park Stacja Wisła’. The study area encompasses objects under conservation policy, i.e. a restaurant ‘Zabłocie 13’ and an independent culture and community centre ‘Warsztat’ (Workshop), which are  included in the context of the new development as representing historic and social values.

[PICTURE]

Refshaleoen in Copenhagen

In 1871, the shipyard of Burmeister & Wain was founded on the island of Refshaleoen. In its peak period it employed several thousand workers. The former icon of the Danish industrial history was closed down in the 1990s, which resulted in the abandonment of many warehouses and factory buildings. Presently, Refshaleoen is a thriving place where its users can spend their free time in an attractive way. Once degraded warehouses and the space between them are now filled with private business, which brought in fresh commercial energy and serves the local community, creating thus a new fashionable district of Copenhagen just 15 minutes away by bicycle from the city centre. A similar distance must be covered to reach the centre of Cracow from the area of Zabłocie. Also, in this case the idea is to create an active fashionable district of the old historic town.

[PICTURE]

Point 2: Taking into account that section 5 is closing the article, the author should use another title (original title for section 5: ‘Summary’.

Response 2: Line 603 the following change was made: “5. Conclusions

Round 3

Reviewer 1 Report

Accept after minor revision (correction of text editing).

Author Response

Response to Comments 6.0

Point 1: The structure of the manuscripts is debatable: For example, the abstract is long-winded, footnotes are unnecessary (they duplicate references in braces), Lines 782-787 better to skip because discredit mentioned PhD-thesis.

Response 1: The structure of the manuscript has been changed according to the Reviewer’s recommendations. Line 13 has been changed in the following way:

Abstract: The subject of this article is the presentation of site conditions and authors’ concept of the development of the degraded riverside area located in the city of Cracow - Kraków Zabłocie. The concept transforms the above-named area into a multifunctional complex including museum, coworking, business and hotel functions. The area subject to development borders three important districts of Cracow: Old Town (Stare Miasto), Grzegórzki and Podgórze on the bank of the Vistula (Wisła) river. In the land development and urban planning documents of the city of Cracow this area has been marked as the public space which is to become a local focal point or a local centre.

The main objective of this work was to find answers to the posed research questions concerning the historic context, formal and legal state, significance for the community as well as economic and ecological implications of the area to be developed. The main purpose was to properly develop the degraded riverside embankment in the downtown environment. The research method was based on own mixed method which encompassed the studies of historical literature and the legal-formal status as well as in-situ examinations, including the analyses of the condition of the built and natural environment, traffic and circulation as well as photographic documentation. Authors also familiarised themselves with the activities undertaken by the local community with a view to the area’s regeneration. On the grounds of initial investigations, the SWOT analysis was performed and the evaluation of groups of prospective users was conducted. Comparative studies were conducted including selected examples of European riverside development projects.

In its assumptions, the proposed concept of the riverside development in Kraków-Zabłocie is to meet the needs of the local community, enable further development of tourism – so important to Cracow, and satisfy the paradigm of sustainable development. The effect is a multi-functional complex which becomes an inherent part of the existing context.

Line 19: Footnote no. [1] was removed and the numbering was changed in the References:

“Cracow Municipal Office - Urząd Miasta Krakowa, A Study of Conditions and Directions of Spatial Development of the City of Cracow (Kraków) - Studium uwarunkowań i kierunków zagospodarowania przestrzennego Miasta Krakowa 2014, Volume 3 Guidelines Referring to Local Development Plans; pp. 29-34, 73-78”

Lines 782-787 were deleted:

This manuscript constitutes a more detailed and extended study of the master’s thesis under the title ‘A Concept of the Development of Podolski Boulevard (Bulwar Podolski) in Kraków Zabłocie into a Multifunctional Complex” written and executed at the Faculty of Architecture of the Silesian University of Technology (Wydział Architektury Politechniki Śląskiej) in 2018, under the supervision of Beata Majerska–Pałubicka, Professor at the Silesian University of Technology, PhD (D.Sc.) Habilitated, Eng., Arch.

Point 2: Problems such as ECOLOGICAL, ECONOMIC, SOCIAL aspects included in the final conclusions (from line 782) do not result from your research. It is better to move them to the DISCUSSION section.

Response 2:

Lines 745-780 were deleted:

“ECOLOGICAL ASPECT

  • In accordance with the directions of development and the management of green areas in Cracow (Kraków), the green character of the study area has been maintained as the area of significant landscape value and ecological value.
  • It is planned to plant a large number of park trees and bushes in the study area. In addition, small architecture and lighting works are planned in the scope required for the users’ safety.
  • Surface area of the biologically active zone was expanded thanks to the design of an underground parking space and parking spaces located along access roads.
  • Study area was linked with the adjacent area of the Boulevard of the Allied Forces Pilots by means of a pedestrian and cycling flyover located above the street of Zabłocie, increasing thus the safety of users.
  • Concept of land development takes into account the design of a new pedestrian and cycling footbridge (linking the districts of Grzegórzki – Podgórze) in order to improve, in an ecological way, the accessibility of the area for new users coming from outside the district of Zabłocie.

ECONOMIC ASPECT 

  • Multifunctional complex features functions satisfying the contemporary needs of users of this fast-transforming district, this fact aims to attract young labour force and increase the attractiveness of this area in relation to other areas located in proximity.
  • Objects under conservation policy: restaurant (Zabłocie 13) as well as culture and community centre ‘Warsztat’ (‘Workshop’ Zabłocie 25) have been included in the development context as objects of historic value and constitute an added value to the developed space by attracting clients.

SOCIAL ASPECT

  • Representative character of the public space was emphasized; it has become a local centre for the local community and visiting tourists.
  • Concept of the multifunctional complex enriches and revitalizes functions connected with culture, recreation and historical education. Being ‘the salon’ of the local centre, the complex with its suitable functions is supposed to attract both local users and visitors having different needs and interests.
  • Multifunctional complex faces the Vistula (Wisła) river with the purpose of creating an attractive panorama of Zabłocie, which can be seen from the opposite river bank, being an important element of the creation of spatial order.
  • Concept of land development takes into account vistas on a local and municipal scale.
  • Study area was developed with the preservation of its unique character – in the form of publicly accessible waterfront of the Vistula River Boulevards (Bulwary Wiślane).”

Line 551, the following text was introduced:

Ecological aspects of the introduced changes are as follows:

  • In accordance with the directions of development and the management of green areas in Cracow (Kraków), the green character of the study area has been maintained in the form of a green area of significant landscape and ecological values.
  • It is planned to plant a large number of park trees and bushes in the study area as well as to implement small architecture and lighting in the scope required for the safety of use.

Line 583, the following text was introduced:

Ecological aspects of the introduced changes are as follows:

  • Reduction of the CO2 emission due to the restriction of motor traffic in favour of cycling.
  • Expansion of biologically-active areas thanks to the design of an underground car park and parking spaces along the access roads.
  • Concept of a new footbridge for pedestrians and cyclists (a cycling link between the districts of Grzegórzki - Podgórze) in order to improve, in an ecological way, the accessibility of the area for new users coming from outside the district.

A social aspect is as follows:

  • Increase in the safety of users by separation of motor traffic from pedestrians and cyclists thanks to a pedestrian and cycling footbridge over the street of Zabłocie, linking the study area with the adjacent areas of the Boulevard of the Allied Forces Pilots.

Line 611, the following text was introduced:

Economic aspects of the introduced changes are as follows:

  • Multifunctional complex features functions satisfying the contemporary needs of users of this fast-transforming district; this fact aims to attract young labour force and increase the attractiveness of this area in relation to other areas located in proximity.
  •  Introduction of workplaces into the district which has a developing residential function will limit the traffic and circulation of inhabitants and thus positively influence the quality of life in the district.
  • Objects under conservation policy: restaurant (Zabłocie 13) as well as culture and community centre ‘Warsztat’ (‘Workshop’ Zabłocie 25) have been included in the development context as objects of historic value and constitute an added value to the developed space by attracting clients.

Social aspects of the introduced changes are as follows:

  • Representative character of the public space was highlighted as a local centre for the district inhabitants and visiting tourists.
  • Concept of the multifunctional complex enriches and revitalizes functions connected with culture, recreation and historical education. Being the ‘salon’ of the local centre, this place and its well-adopted functions are expected to attract users and visitors with different needs and interests.
  • Multifunctional complex faces the Vistula (Wisła) river in order to create an attractive panorama of Zabłocie, which can be seen from the opposite riverbank; this constitutes an important element of the formation of spatial order.
  • Concept of land development takes into account the existing vistas on a local and urban scale.
  • Study area was developed with the preservation of its unique, on a European scale, character – in the form of a commonly accessible waterfront space called the Vistula Boulevards (Bulwary Wiślane).

Point 3: Your comments on analysis and synthesis are trivial. They don't explain the research method you use. After all, the reviews asked for an explanation of your method! In the abstract you even suggest the heuristic method; you further limit yourself to a laconic mention of this method.

Response 3:

[ Both basic methods (the method of logical argumentation and the heuristic method) and own mixed method were described in previous corrections introduced to the text in accordance with the Reviewers’ recommendations. Basic methods were on purpose described in a smaller extent than own mixed method because the latter constitutes the most important element, which was used in further research. ]

Line 90, was changed:

One of the fundamental assumptions of this work is a holistic multifaceted analysis of the study area. In order to make the most appropriate decisions concerning the selection of a research method, the authors analysed the suitability of a series of research methods usually applied to complex issues requiring an extensive analysis. The study was based on two, in the authors’ opinion the most adequate, research methods in a given context, namely:

  • Method of logical argumentation – as a search for theoretical interpretation of developments (events) with the application of a logical description of reality, based on analysis and synthesis.
  • Heuristic method – understood as ways and rules of proceedings serving the purpose of making the most appropriate decisions in complicated situations, requiring the analysis of available information and the prediction of future phenomena. The method is based on creative thinking and logical combinations [6].

[ Below please find a broader  description of the two methods which constituted the basis for our own mixed method: ]

Both the method of logical argumentation and heuristic methods help to systemize the activities connected with investigations, beginning from the determination of the study subject and ending at the definition of expected results from the work conducted. In the above-mentioned methods, the subject and scope of the study defined aspects such as the search for key developmental factors, types of investigations (theoretical, analytical), humanistic and philosophical interpretation of architectural issues, etc. Next, the types of undertaken activities are determined (deduction, synthesis, analogies, drawing conclusions, abstract and logical analysis). The heuristic methodology specifies a study approach, for instance: if it is expected to share one’s knowledge and opinions as well as if the researcher should enter some social interactions. Both methods also define techniques applied in order to carry out investigations, such as: logical interpretations, SWOT analyses, scenarios, Delphi technique, foresight. The researcher also determines which tools will be used (architectural documentation, lists and comparisons, tables). Finally, the expected results are predicted. In the heuristic method: determination of possible directions of development (empirical approach), definition of methods of implementation of strategic goals (normative approach), or in the case of the method of logical argumentation, for example: if it is planned to publish the study results.

[ Investigation method (own mixed method) applied is explained below as the transformation of two previous methods into own mixed method: ]

The combination of two methods into one mixed method aims at the systematization of the undertaken work which is supposed to yield specific effects. While creating own mixed method the authors based it on the rules and methods used in deconstructivism, which challenges fixed patterns and ponders on a given issue anew, from scratch, following simultaneously technological changes. This approach was very helpful when it came to the questioning of the legitimacy of the so-far well-established, either by law or custom, decisions and activities. It facilitated also the search for the balance between individual issues, which had been earlier deconstructed into separate elements. In Tischner’s book we read that: ‘Jackques Derrida proposed that the act of creation should be the goal in its own right, not a piece of creation itself, even in architecture. However, this seems to be an extreme view, suspending the centuries-long aim of building engineering’ [7].
